# Abnormal degradation of the neuronal stress-protective transcription factor HSF1 in Huntington's disease

Rocio Gomez-Pastor[1], Eileen T. Burchfiel[1,2], Daniel W. Neef[1], Alex M. Jaeger[1], Elisa Cabiscol[3], Spencer U. McKinstry[4], Argenia Doss[5], Alejandro Aballay[5], Donald C. Lo[6], Sergey S. Akimov[7], Christopher A. Ross[7,8,9,10], Cagla Eroglu[4,6] & Dennis J. Thiele[1,2,5]

Huntington's Disease (HD) is a neurodegenerative disease caused by poly-glutamine expansion in the Htt protein, resulting in Htt misfolding and cell death. Expression of the cellular protein folding and pro-survival machinery by heat shock transcription factor 1 (HSF1) ameliorates biochemical and neurobiological defects caused by protein misfolding. We report that HSF1 is degraded in cells and mice expressing mutant Htt, in medium spiny neurons derived from human HD iPSCs and in brain samples from patients with HD. Mutant Htt increases CK2α' kinase and Fbxw7 E3 ligase levels, phosphorylating HSF1 and promoting its proteasomal degradation. An HD mouse model heterozygous for CK2α' shows increased HSF1 and chaperone levels, maintenance of striatal excitatory synapses, clearance of Htt aggregates and preserves body mass compared with HD mice homozygous for CK2α'. These results reveal a pathway that could be modulated to prevent neuronal dysfunction and muscle wasting caused by protein misfolding in HD.

[1] Department of Pharmacology and Cancer Biology, Duke University School of Medicine, Durham, North Carolina 27710, USA. [2] Department of Biochemistry, Duke University School of Medicine, Durham, North Carolina 27710, USA. [3] Departament de Ciencies Mediques Basiques, IRB Lleida, Universitat de Lleida, Lleida 25008, Spain. [4] Department of Cell Biology, Duke University School of Medicine, Durham, North Carolina 27710, USA. [5] Department of Molecular Genetics and Microbiology, Duke University School of Medicine, Durham, North Carolina 27710, USA. [6] Department of Neurobiology, Duke University School of Medicine, Durham, North Carolina 27710, USA. [7] Division of Neurobiology, Department of Psychiatry, Johns Hopkins University School of Medicine, Baltimore, Maryland 21205, USA. [8] Department of Neurology, Johns Hopkins University School of Medicine, Baltimore, Maryland 21205, USA. [9] Department of Pharmacology, Johns Hopkins University School of Medicine, Baltimore, Maryland 21205, USA. [10] Department of Neuroscience, Johns Hopkins University School of Medicine, Baltimore, Maryland 21205, USA. Correspondence and requests for materials should be addressed to D.J.T. (email: dennis.thiele@duke.edu).

Huntington's Disease (HD) is a genetically encoded autosomal dominant neurodegenerative disease caused by a poly-glutamine (Q) expansion (CAG trinucleotide repeat) within exon 1 of the Huntingtin (HTT) gene[1]. The encoded Htt-polyQ protein is expressed in almost all cells, which leads to defects in transcription, autophagy, mitochondrial function, signalling and apoptosis[2,3]. Although HD preferentially affects neuronal function and the survival of striatal and cortical neurons, defects are observed in peripheral tissues in mouse models and in patients that include skeletal muscle wasting and cardiac atrophy, perhaps reflecting toxicity and apoptosis due to the ubiquitous expression of Htt-polyQ (ref. 4).

The presence of a pathogenic polyQ expansion causes Htt to misfold and aggregate, driving inappropriate interactions with transcription factors, signalling and cell integrity proteins and other key cellular regulatory factors in both the cytosol and nucleus[5]. The protein quality control machinery, including chaperones, the ubiquitin proteasome, autophagy and other factors play critical roles in the folding, trafficking, modification and degradation of both newly synthesized and misfolded proteins in disease[6,7]. Accordingly, increased expression of chaperones such as Hsp104, Hsp70, Hsp40 and Hsp27, or critical components in the autophagy pathway, ameliorates protein aggregation and cell death in cellular, fly, worm and mouse polyQ expansion disease models[8–11]. As chaperones function in obligate hetero-multimeric complexes, the coordinate expression of distinct chaperones synergize in the amelioration of polyQ protein aggregation and cellular stress protection in polyQ-expansion models[12].

Heat shock transcription factor 1 (HSF1) is a stress-responsive transcription factor that protects cells from protein misfolding, aggregation and apoptosis[13] by expressing genes involved in protein quality control, stress adaptation and cell survival[14]. HSF1 is activated in response to elevated temperature, oxidant exposure, metals and other conditions that cause protein misfolding[15]. Under normal cell growth conditions, HSF1 is present as an inactive monomer repressed by Hsp40, Hsp70, Hsp90 and TRiC, protein chaperones involved in the folding and maturation of hundreds of cellular client proteins[16–18]. In response to proteotoxic stress HSF1 assembles as a multimer, binds heat shock elements in target gene promoters and activates expression of stress-protective genes[19]. HSF1 undergoes many post-translational modifications including both basal and stress-induced phosphorylation, sumoylation, ubiquitinylation and acetylation that mediate repressive or activating regulatory roles[20–22].

Consistent with HSF1 activating protein folding and stress-protective pathways, hsf1$^{-/-}$ mice in the context of an R6/2 HD model show increased brain Htt aggregation and a shortened lifespan[23], while expression of a constitutively active form of HSF1 inhibited Htt-polyQ aggregation and prolonged lifespan[24]. Moreover, a heterozygous HSF1 mouse model of spinal and bulbar muscular atrophy with a pathogenic polyQ repeat in the androgen receptor (AR), exhibited increased AR-polyQ aggregates in neurons and non-neuronal tissues and enhanced neurodegeneration[25,26].

While there is strong evidence for beneficial effects of HSF1 in polyQ expansion models, HSF1 target gene expression is compromised in the presence of disease-associated polyQ-expansion proteins[27–30]. Pharmacological activation of HSF1 with a blood–brain barrier-penetrant Hsp90 inhibitor increased HSF1 target gene expression and was initially effective in disease amelioration[28]. However, this beneficial effect was observed only at early stages and was proposed to be due to the inability of HSF1 to bind target genes in the altered chromatin environment found in the R6/2 mouse model. Other reports suggest that HSF1 protein levels may affect the expression of the protein folding machinery components in HD models[29,31]. Given the therapeutic potential for HSF1 activation in protein misfolding disease[32–34], it is important to clarify our understanding of the mechanisms by which HSF1 activation is defective in HD.

Here we demonstrate that HSF1 protein levels are strongly decreased in HD models, in differentiated human inducible pluripotent stem cells and in HD patient striatum and cortex, with a concomitant defect in target gene expression. This defect is due to inappropriate degradation of HSF1 via phosphorylation-stimulated ubiquitin-dependent degradation induced by abnormally high levels of the casein kinase 2α (CK2α)-prime (CK2α′) kinase and the Fbxw7 E3 ligase. Diminution of CK2α′ expression in the zQKI175 mouse HD model restores HSF1 levels and activity and prevents mutant Htt aggregation, striatal excitatory synapse loss and cachexia. These studies elucidate a critical molecular mechanism for inappropriate degradation of the protein misfolding stress-protective transcription factor HSF1 and suggests a novel therapeutic target (CK2α′) that is potentially amenable to pharmacological intervention for the treatment of HD.

## Results

**Decreased HSF1 correlates with elevated HSF1-P-S303/307.** To explore whether the defect in HSF1 activation lies in changes in HSF1 abundance in the presence of misfolded polyQ-Htt, we compared HSF1 protein levels, target promoter occupancy and gene expression in PC12 cells with Tetracycline inducible cassettes expressing human Htt exon 1 with non-pathogenic Q repeats (Htt-Q23-green fluorescent protein (GFP)) versus pathogenic repeats (Htt-Q74-GFP). We performed these studies under control (37 °C) and heat shock conditions (42 °C), allowing us to both analyse the defect in HSF1 under polyQ protein expressing conditions and explore the expression of inducible HSF1 target genes (Fig. 1a). The HSF1 targets Hsp70, Hsp25, Hspb5 and Bag3 are strongly expressed in response to heat shock activation in Htt-Q23 cells and in Htt-Q74 cells in the absence of tetracycline ( − Dox), while Htt-Q74-expressing cells ( + Dox) showed a strong decrease in their expression in response to heat shock (Fig. 1b; Supplementary Fig. 1A,B). We also analysed Hsp70 and Hsp25 transcript levels over time after heat shock in Htt-Q74 cells (Fig. 1c). The decrease in HSF1 target gene expression in Htt-Q74-expressing cells (Fig. 1b and Supplementary Fig. 1A) correlated with decreased Hsp70 and Hsp25 messenger RNA (mRNA) levels (Fig. 1c) and a parallel reduction in HSF1 occupancy of the Hsp70 promoter (Fig. 1d). A similar defect in HSF1 activation was observed in response to azetidine 2-carboxylic acid, a proline analogue that causes protein misfolding and HSF1 activation (Supplementary Fig. 1C). These results demonstrate a clear defect in HSF1 DNA binding and target gene activation under protein misfolding stress conditions in Htt-Q74 cells, but not Htt-Q23 cells.

A reduction in HSF1 protein levels in Htt-Q74-expressing cells was observed under control conditions that was exacerbated in response to proteotoxic heat shock conditions (Fig. 1b). A similar reduction was observed using whole-cell extracts prepared with urea or acetone precipitation, ruling out the possibility of HSF1 partitioning into the insoluble fraction (Supplementary Fig. 1D,E). However, no significant changes in HSF1 mRNA were observed (Supplementary Fig. 1F). This reduction in HSF1 levels was accompanied by increased HSF1 phosphorylation at Ser303 and Ser307, located within the central regulatory domain[35–37]. Induction of Htt-Q74 expression greatly elevated S303/307 phosphorylation, further enhanced in response to heat shock (Fig. 1b), and occurred in a time-dependent fashion

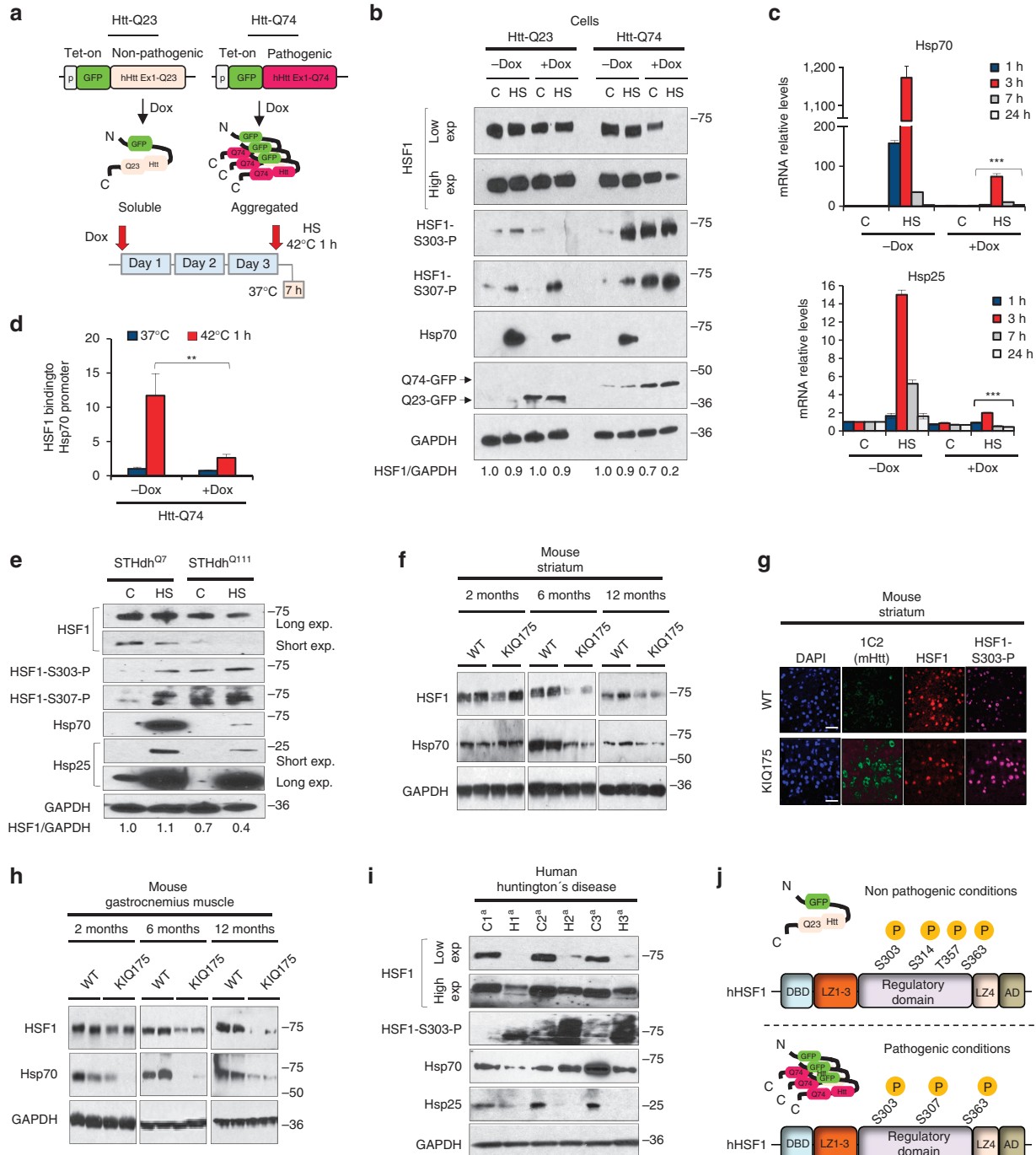

**Figure 1 | HSF1 levels are decreased and P-Ser303/307 increased in HD.** (**a**) Diagram of the PC12-HttQ23 and Htt-Q74 proteins and experimental design. (**b**) Cells expressing either Htt-Q23 or Htt-Q74 were cultivated at 37 °C (**c**) in the presence of tetracycline (Dox) for 3 days to induce expression, exposed to heat shock 1 h at 42 °C, allowed to recover at 37 °C for 7 h (HS) and protein extracts immunoblotted with the indicated antibodies. HSF1 bands were quantitated using Quantity One image software (BioRad) and values normalized using GAPDH as loading control and referenced to control at 37 °C (**c**) in the absence of −Dox. (**c**) qRT-PCR analysis of the Hsp70 and Hsp25 genes in Htt-Q74-expressing cells as in B. HS (+Dox) group was compared with HS (−Dox) group. Error bars represent means ± s.e.m., $n = 4$. Unpaired $t$-test, ***$P < 0.001$. (**d**) Analysis of Hsp70 promoter occupancy by HSF1. Error bars represent means ± s.e.m., $n = 3$. Unpaired $t$-test, **$P < 0.01$. (**e**) Mouse-derived striatal STHdh$^{Q7}$ and STHdh$^{Q111}$ cells were cultured at 33 °C (**c**), heat shocked at 42 °C 1 h with recovery at 33 °C for 7 h and immunoblotted with the indicated antibodies. HSF1 was quantified using Quantity One image software and normalized using GAPDH as loading control and referenced to control at 37 °C in the non-pathogenic STHdh$^{Q7}$. (**f**) Striatal samples from Wild type (WT) C57BL/6 or KIQ175 mice at 2, 6 and 12 months analysed by immunoblotting ($n = 4$). (**g**) Dorsal striatal sections from WT and KIQ175 mice ($n = 3$) at 6 months assayed by immunohistochemistry (IHC) for mHtt aggregation (1C2), HSF1 and HSF1-S303 phosphorylation using DAPI staining as control. Scale bar: 10 μm. (**h**) Gastrocnemius muscle extracts from WT and KI175 mice ($n = 4$) of the indicated age, immunoblotted for HSF1, Hsp70 and GAPDH. (**i**) Protein extracts from HD patient striatum (Supplementary Table 1) and controls immunoblotted with indicated antibodies. (**j**) HSF1 phosphoproteomic analysis under non-pathogenic (−Dox) and pathogenic (+Dox) conditions in hsf1$^{−/−}$ MEF inducible cell line expressing GFP-Htt-Q74 expressing HSF1. See Supplementary Fig. 8 for uncropped immunoblots. HSF1 represented by the Regulatory domain, DBD; DNA binding domain; LZ1-3 and LZ4; leucine zipper domains.

(Supplementary Fig. 1G). These same changes were observed in the striatal neuron-derived cells from STHdh$^{Q111/Q111}$ knock-in mice when compared with the non-pathogenic STHdh$^{Q7/Q7}$ cells (Fig. 1e). A spinal and bulbar muscular atrophy PC12 cell model (AR-Q112), also showed decreased HSF1 and Hsp25/Hsp70 expression levels and increased HSF1-S303 phosphorylation compared with non-pathogenic cells (AR-Q10; (Supplementary Fig. 1H). Together, distinct cellular models of pathogenic polyQ-expansion protein misfolding disease show defective HSF1 activation, increased HSF1 S303/307 phosphorylation and decreased HSF1 protein levels.

To ascertain if these observations made in cellular models of HD are found in animal models of HD, HSF1 levels were evaluated in wild-type (WT) C57Bl/6 mice and in the zQ175 (herein KIQ175) heterozygous knock-in HD model. This mouse model exhibits striatal and cortical excitatory synaptic defects, abnormal gait and motor defects, loss of body weight, reduced levels of striatal marker genes and other defects similar to human HD patients[38,39]. Although striatal tissue from 2-month-old KI175 mice showed little difference in the levels of HSF1 compared with wild-type littermates, a strong reduction in HSF1 levels was observed at both 6 and 12 months, as was a reduction in Hsp70 (Fig. 1f). This correlates with the appearance of Htt-polyQ aggregates, increased HSF1 S303 phosphorylation in the striatum (Fig. 1g and Supplementary Fig. 1I) and the age of onset of motor deficits in this HD model[38,39]. Similar results were observed in the cortex and gastrocnemius muscle, where HSF1 and Hsp70 dramatically decreased at 6 and 12 months of age (Fig. 1h and Supplementary Fig. 1J). Interestingly, Hsp70 levels in the gastrocnemius muscle of KI175 HD mice also showed a strong diminution from that found in wild-type mice at 2 months, even before a strong reduction in HSF1 levels (Fig. 1h). These results demonstrate that HSF1 levels are reduced in striatal and cortical tissues, and in skeletal muscle in the heterozygous KI175 HD model, even before the reported onset of motor or cognitive deficits[38,39].

To evaluate HSF1 levels in patients with HD, post-mortem age- and sex-matched HD and control striatal and cortex samples were assessed using a total of 14 independent striatal samples and 7 cortex samples (Fig. 1i; Supplementary Fig. 2A–C,F,G and Supplementary Table 1). In 10 of 14 samples analysed from the striatum of HD patients, and 5 of 7 samples analysed from the cortex, a strong reduction in HSF1 levels was evident. Moreover, HD protein extracts displayed reduced HSF1 electrophoretic mobility, irrespective of residual protein levels and tissue sample source, and an increase in HSF1 S303 phosphorylation (Fig. 1i). Accordingly, the levels of Hsp70 and Hsp25 were decreased in 7 out of 10 HD samples that exhibited decreased HSF1 relative to controls (Fig. 1i; Supplementary Fig. 2A,B,F), consistent with a previous transcriptome analyses of human HD brain[40].

To explore the role of S303 and S307 phosphorylation in HSF1 regulation in HD, HSF1 phosphoproteomics was performed in hsf1$^{-/-}$ MEFs inducibly expressing human Htt-Q74 or Htt-Q23 and transfected with an HSF1 expression vector (Fig. 1j and Supplementary Data 1). Out of 95 total serine and threonine residues present in HSF1, only four were detected as phosphorylated in Htt-Q23 cells corresponding to S303, S314, T357 and S363. In Htt-Q74 cells HSF1 was phosphorylated at S303, S307 and S363, highlighting the increased occurrence of HSF1 S307 phosphorylation under pathogenic polyQ conditions. Although this analysis does not quantify the abundance of phosphorylation events, these and other results (Fig. 1b,e,g,i and Supplementary Fig. 1G) demonstrate that HSF1 S303 and S307 phosphorylation are specifically increased in the presence of a pathogenic polyQ protein in parallel with decreased HSF1 protein abundance and activity.

**Htt-polyQ-dependent HSF1 degradation by the Fbxw7 E3 ligase.** HSF1 protein levels and Hsp70 expression are progressively reduced when Htt-Q74 is expressed in PC12 cells over the course of 3 days (Fig. 2a,b), with little difference in the steady-state levels of HSF1 mRNA (Supplementary Fig. 1F), correlating with increased HSF1-S303 phosphorylation (Fig. 2a,b). We tested whether Htt-Q74 expression stimulates HSF1 degradation by evaluating HSF1 protein levels and activation in the presence of the proteasome inhibitor MG132, which also results in the accumulation of misfolded proteins and activation of HSF1. Cells not expressing Htt-Q74 displayed robust activation of Hsp70 in the presence of either MG132, or the HSF1 activator 17-AAG, which inhibits Hsp90 function (Fig. 2c,d). In contrast, the same cells expressing Htt-Q74 were defective in Hsp70 expression when treated with 17-AAG and exhibited decreased HSF1 levels (Fig. 2c). As an indication of HSF1 proteasomal degradation, both HSF1 levels and activity were preserved by co-incubation with 17-AAG and MG132. Under the latter condition, HSF1 migrated through SDS–polyacrylamide gel electrophoresis (PAGE) as multiple species (Fig. 2c), suggestive of one or more HSF1 post-translational modifications. To test whether Htt-Q74 induces ubiquitin-dependent proteasomal degradation of HSF1, PC12 cells were transfected with a plasmid expressing HA-tagged ubiquitin, induced to express Htt-Q74, treated with MG132, and cell extracts or HSF1 immuno-precipitates immunoblotted (Fig. 2d,e). Both HSF1 protein levels and the HSF1-S303 phosphorylated species were increased in the presence of MG132 as a consequence of blocking the proteasome, and higher molecular weight HSF1-ubiquitylated forms were observed when Htt-Q74 was expressed in the presence of MG132, consistent with ubiquitin-proteasome-dependent degradation (Fig. 2d,e).

The SCF-$^{FBW}$ (Skp1-Cul1-F box) ubiquitin ligase complex protein Fbxw7 ubiquitinylates HSF1 in cancer cells in a S303/307 phosphorylation-dependent manner[41]. In melanoma cells Fbxw7 abundance is decreased, resulting in increased HSF1 protein levels. The analysis of mRNA levels of three E3 ligases (CHIP, HECTD and Fbxw7) in the striatum of WT and KIQ175 mice revealed increased Fbxw7 in the KIQ175 mice (Fig. 2f). Increased Fbxw7 protein levels were observed in STHdh$^{Q111}$ cells, KIQ175 mice and HD patients (Fig. 2g–i; Supplementary Fig. 2A,B,E) and correlated with increased HSF1 S303/307 phosphorylation and low HSF1 protein levels and activity (Figs 1b–f,i and 2e; Supplementary Fig. 1G). Knocking down Fbxw7 in STHdh$^{Q7}$ and STHdh$^{Q111}$ cells increased HSF1 levels in both cells, with the increase more pronounced in STHdh$^{Q111}$ cells, suggesting a role for this E3 ligase in HSF1 degradation in HD (Fig. 2i).

A previous report showed that the interaction between HSF1 and Fbxw7 is diminished with HSF1 S303A and/or S307A mutants in cancer cells[41]. Indeed, co-immunoprecipitation experiments show that HSF1 S303A mutant interacts less robustly with Fbxw7 compared to WT HSF1 when expressed in hsf1$^{-/-}$ MEFs (Fig. 2j), recapitulating a role for this phosphorylation event in the interaction of HSF1 with Fbxw7. Given that HSF1 S303A exhibited a reduced interaction with Fbxw7, the effect of the HSF1-S303A mutation on HSF1 stability was assessed by expressing WT HSF1 or HSF1 S303A in hsf1$^{-/-}$ MEFs expressing Dox-inducible Htt-Q74 (Fig. 2k). As expected, hsf1$^{-/-}$ MEFs expressing Htt-Q74 exhibit decreased HSF1 protein levels and activity and increased S303 phosphorylation, as compared to un-induced cells. However, the HSF1 S303A mutation, which precludes phosphorylation at this site, increased HSF1 protein levels and Hsp70 expression in the presence of Htt-Q74 protein (Fig. 2k). These results suggest that S303 phosphorylation is critical for Htt-polyQ-dependent degradation of HSF1 via the action of Fbxw7 and the proteasome.

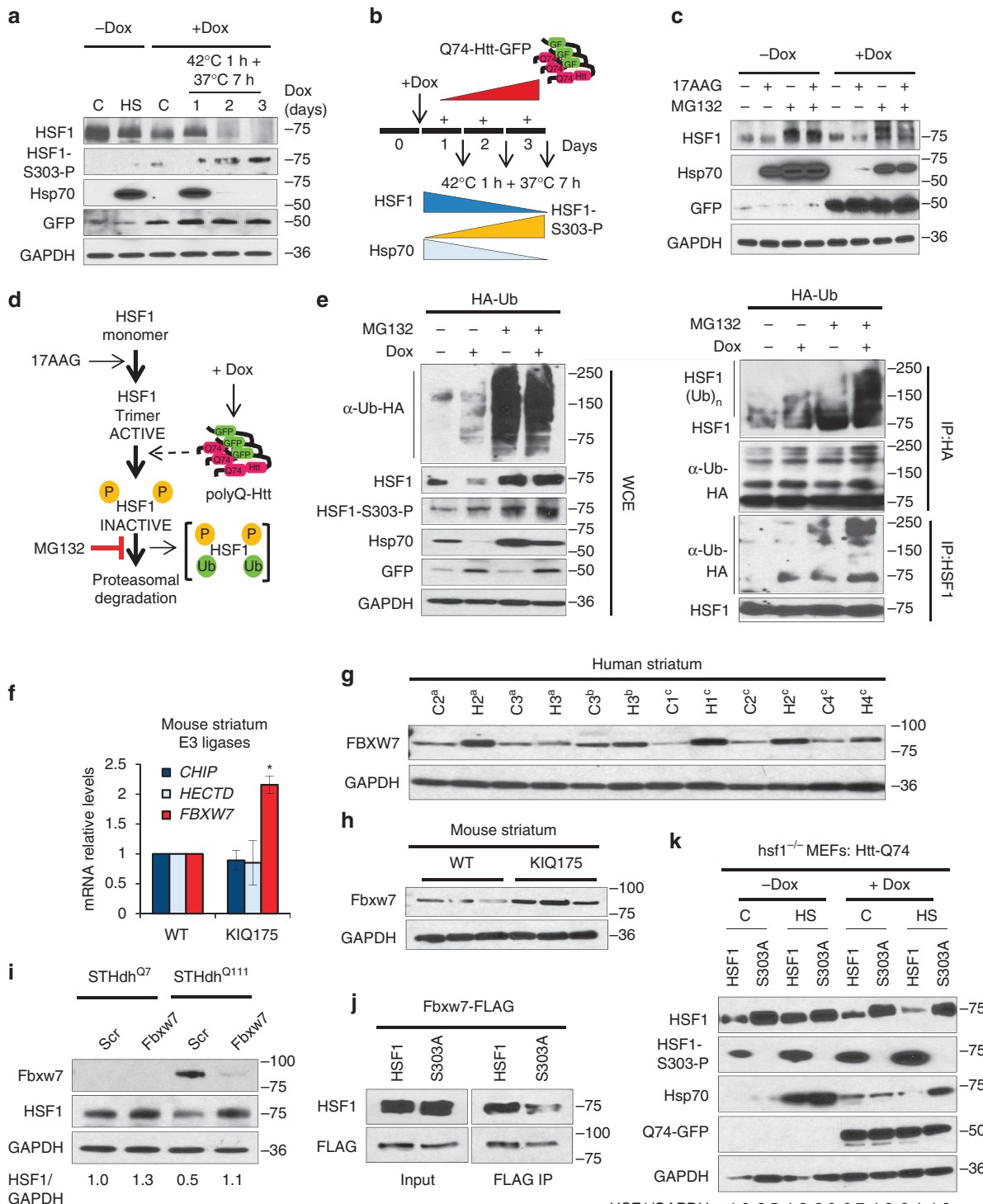

**Figure 2 | Proteasomal HSF1 degradation in HD is mediated by Phospho-S303/S307. (a,b)** PC12 cells expressing Htt-Q74 for 1, 2 or 3 days followed by Heat Shock (HS) and recovery as indicated. Control and HS in the absence of Dox correspond to cells incubated during 3 days at 37 °C. (**c**) Htt-Q74 cells Dox-induced or not for 3 days and exposed to 2 μM 17-AAG and/or MG132 (5 μM) for 6 h and extracts immunoblotted with the indicated antibodies. (**d**) Diagram of the effects of 17-AAG and MG132 treatment and Htt-Q74 expression in HSF1. (**e**) Htt-Q74 cells transfected with human influenza haemagglutinin-ubiquitin (HA-Ub) plasmid Dox induced or not and treated with 5 μM MG132 treatment for 6 h. Whole-cell extract and HA immunoprecipitated (IP:HA) and HSF1 immunoprecipitated samples (IP:HSF1) were immunoblotted as indicated. (**f**) Transcript levels for indicated E3 ligases evaluated by qRT–PCR from striatum of WT and KIQ175 mice at 6 months. Error bars represent ± s.e.m., ($n = 3$). Unpaired $t$-test, *$P < 0.05$. (**g**) Human striatum samples from HD patients and controls and (**h**) Mouse striatum from 12 months old WT and KIQ175 mice were immunoblotted for HSF1 and Fbxw7. (**i**) Fbxw7 siRNA in STHdh$^{Q7}$ and STHdh$^{Q111}$ cells using scrambled RNA (Scr) as control. HSF1 was quantified using Quantity One image software and normalized using GAPDH as loading control and referenced to control at 37 °C in the non-pathogenic STHdh$^{Q7}$. (**j**) hsf1$^{-/-}$ MEFs transfected with WT HSF1 or S303A mutant and Fbxw7-FLAG; samples immunoprecipitated with anti-FLAG and HSF1 detected. (**k**) hsf1$^{-/-}$ MEFs expressing Dox-inducible Htt-Q74-GFP transfected with WT HSF1 or HSF1-S303A HS and 1 h recovery at 37 °C for 7 h. HSF1 was quantified using Quantity One image software and normalized using GAPDH as loading control and referenced to control ( − Dox) expressing WT HSF1. See Supplementary Fig. 9 for uncropped immunoblots.

**Htt-polyQ-dependent HSF1 phosphorylation by Casein Kinase 2.**
To identify protein kinases involved in controlling HSF1 phosphorylation and degradation in HD, we used a previously developed screen for regulators of human HSF1 expressed in yeast *hsf1Δ* cells (Fig. 3a)[42]. The basis of the screen is that human HSF1 (hHSF1) exists in an inactive form in yeast and does not complement the viability defect of *hsf1Δ* cells. Yeast *hsf1Δ* cells harbouring both a galactose-inducible, glucose-repressible yHSF1 plasmid and a constitutively expressed human HSF1 plasmid were cultured in galactose conditions to allow yHSF-dependent growth. In glucose medium human HSF1 is insufficient to promote viability unless small molecules activate HSF1 and promote cell growth (Fig. 3a). A library of commerical Ser/Thr and Tyr protein kinase inhibitors was evaluated for molecules able to support cell viability in a human HSF1-dependent manner (Fig. 3b). All five hits from this screen have the common feature that, among other kinases, they inhibit CK2, a kinase in which the holoenzyme is composed of 2 regulatory and 2 catalytic subunits[43,44] (Fig. 3c). These results suggested an inhibitory role for CK2 in human HSF1 regulation in yeast. Because of the limited specificity of these kinase inhibitors, the modulation of human HSF1 by yeast CK2 was tested by deletion of the *CKB1* gene, encoding a CK2 stimulatory subunit in *S. cerevisiae* (Fig. 3c). When the yeast HSF1 expression plasmid was evicted from cells expressing human HSF1, isogenic *ckb1Δ* cells were viable and exhibited increased human HSF1 protein levels compared with WT cells (Fig. 3d–f).

*In vitro* phosphorylation assays were conducted using purified recombinant human glutathione S-transferase (GST)- CK2α or CK2α′ catalytic subunits, or commercial CK2 holoenzyme, and recombinant human HSF1 (Fig. 3g). Mass spectrometry analysis of HSF1 revealed that 11 Ser and 6 Thr residues were phosphorylated *in vitro* by CK2 (Fig. 3g and Supplementary Data 2) with most located in the regulatory domain. Although some residues were phosphorylated by all three CK2 preparations, other phosphorylation sites were unique. This is particularly relevant since the catalytic subunits CK2α and CK2α′ can function *in vivo* in the absence of the CK2β regulatory subunit[43]. Intriguingly, HSF1-S303-P was identified only in the presence of CK2 holoenzyme and HSF1-S307-P was identified solely in the presence of CK2α′. Of the HSF1 residues phosphorylated *in vitro* by CK2, 8 different Ser and Thr residues were mutated to encode Ala (Supplementary Fig. 3A) but only the HSF1-S303A and/or S307A mutants promoted robust human HSF1-dependent yeast growth (Fig. 3h and Supplementary Fig. 3B) consistent with previous studies of these mutants showing elevated HSF1 protein levels in yeast[32]. The HSF1-S303A and S303A/S307A mutants were further evaluated in the humanized yeast assay in the presence of the CK2 inhibitor TID43. While TID43 administration allowed WT human HSF1-dependent yeast growth and potentiated S303A mutant growth, no further growth stimulation was observed for the S303/307A double mutant, showing insensitivity to pharmacological CK2 inhibition (Fig. 3h). Taken together these results demonstrate that CK2 directly phosphorylates human HSF1 at S303 and S307 *in vitro* and pharmacological or genetic inhibition of these events increases human HSF1 activity and abundance in yeast.

To investigate a role for mammalian CK2 (Fig. 4a) in HSF1 degradation in HD, Htt-Q74 expressing cells were incubated with two CK2 inhibitors that activate human HSF1 in the yeast assay, TID43 and Emodin (Fig. 3b). Both molecules facilitated Hsp70 expression in non-pathogenic (Htt-Q23) and pathogenic (Htt-Q74) conditions in a dose-dependent manner (Fig. 4b,c) and activated HSF1 in human ARPE cells (Supplementary Fig. 4A,B). Incubation of Htt-Q74 cells with TID43 or Emodin before or after Htt-Q74 induction (Supplementary Fig. 4C)

decreased HSF1-S303 phosphorylation, increased HSF1 and Hsp70 levels (Fig. 4d and Supplementary Fig. 4D), decreased Htt-Q74 aggregates (Fig. 4e,f; Supplementary Fig. 4E) and increased Hsp70 expression and cell viability in an HSF1-dependent manner (Fig. 4g and Supplementary Fig. 4F,G). While Gsk3β phosphorylates HSF1 S303 and drives HSF1 degradation in cancer cells[36,41], a GSK3 inhibitor or GSK3 RNAi knockdown moderately stimulated Hsp70 expression (Supplementary Fig. 4E,H) and did not impact Htt-Q74 aggregation (Supplementary Fig. 4E). Furthermore, knockdown of Gsk3α and/or Gsk3β had a detrimental effect on HSF1 protein stability and did not change the HSF1-S303 phosphorylation state (Supplementary Fig. 4H).

While yeast CK2 catalytic activity is positively regulated by the CKB subunits (Fig. 3c), mammalian CK2 activity can be either activated or inhibited by the CK2β regulatory subunit (Fig. 4a) thus modulating activity towards specific substrates[43,44]. Consistent with CK2β functioning as a negative regulator of HSF1 phosphorylation by CK2, silencing of CK2β in the presence of Htt-Q74 decreased HSF1 protein levels and Hsp70 expression (Fig. 4h). However, short interfering RNA (siRNA)-mediated knockdown of the catalytic CK2α′ subunit alone, or in combination with the CK2α subunit, reduced HSF1 S303 and S307 phosphorylation (Fig. 4i), increased HSF1 protein levels and Hsp70 expression and increased cell viability in the presence of Htt-Q74 (Fig. 4i; Supplementary Fig. 4I–K). A knockdown of *kin-3*, the sole *C. elegans* CK2 catalytic subunit (Supplementary Fig. 4L), resulted in higher expression of *hsp-25* in an HSF1-dependent manner, suggesting a conservation of function, but it did not suppress protein aggregation or toxicity in animals overexpressing polyQ::YFP fusion proteins in body wall muscle cells, perhaps due to the potent toxicity of the Q37::YFP model (Supplementary Fig. 4M).

**CK2α′ is highly abundant in HD medium spiny neurons.** The abundance of the CK2 catalytic and regulatory subunits was evaluated in cell and mouse HD models and in striatum from patients with HD (Fig. 5). In Htt-Q74 expressing PC12 cells CK2α′ showed increased abundance that was further increased under HS conditions (Fig. 5a), correlating with increased HSF1 S303/307 phosphorylation and decreased HSF1 levels (Fig. 1b). In the KIQ175 HD mouse model CK2α′ mRNA (Fig. 5b) and protein levels were markedly increased in striatum and gastrocnemius muscle (Fig. 5c,d). Immuno-histochemical analysis of the dorsal striatum also revealed that CK2α′ expression was increased in KIQ175 mice (Fig. 5e) while no signal was detected in CK2α′−/− mice (Supplementary Fig. 5A). CK2α′ expression co-localized with the neuronal marker (NeuN) and the medium spiny neuron (MSN) markers Darpp32, Ctip2 and Fox1P (refs 45,46) in both WT and KIQ175 mice (Fig. 5e and Supplementary Fig. 5B,C). The analysis of the GS and CD68 markers showed no co-localization with astrocytes or reactive microglia, respectively (Supplementary Fig. 5B,D). These results demonstrated that MSNs express increased CK2α′ in the presence of mHtt (Supplementary Fig. 5E). Strikingly, striatal tissue from patients with HD also exhibited elevated CK2α′ mRNA and protein levels (Fig. 5f,g). Nine of 14 patients with HD showed increased CK2α′ expression in the striatum compared with control samples and among all 10 patients that showed reduced HSF1, 7 showed increased CK2α′ (Supplementary Fig. 2A,B,D).

**HSF1 is degraded in mouse and human MSN-like HD cells.** Our findings demonstrate a correlation between decreased HSF1 abundance and activity, and increased expression of the HSF1 degradation machinery components CK2α′ and Fbxw7. These

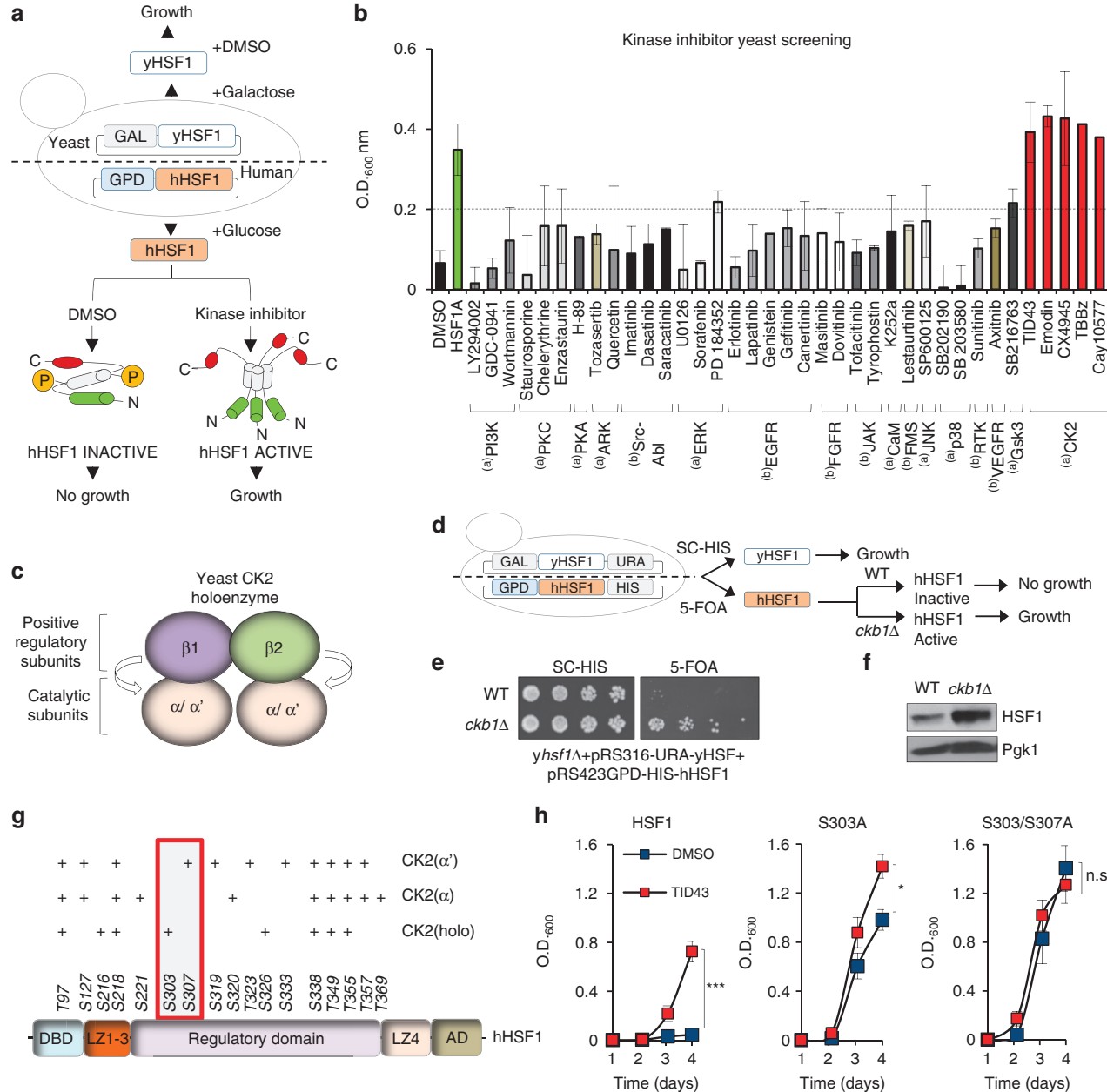

**Figure 3 | CK2 kinase modulates human HSF1 activity and stability in yeast.** (**a**) Experimental design of humanized HSF1 yeast screen for kinase inhibitors that promote yeast HSF1-dependent growth. (**b**) Yeast cells expressing human HSF1 cultivated with ser-thr kinase inhibitors and growth (OD$_{600}$ nm) monitored over 4 days. Data presented correspond to results at 20 nM concentration. Similar results were obtained at other tested drug concentrations from 2 nM to 20 μM. DMSO was negative control and HSF1A used as positive control. (a = Ser/Thr kinases, b = Try kinases). Error bars represent ± s.e.m., (n = 4). (**c**) Yeast CK2 holoenzyme subunit composition and function. (**d**) Experimental design of humanized HSF1 yeast screen for *CKB1* deletion that promote yeast HSF1-dependent growth. (**e**) WT and *CKB1* mutant strain (*ckb1Δ*) grown in SC-His or 5-FOA medium for 3 days at 30 °C. (**f**) Protein extracts from WT (*CKB1*) and mutant strain (*ckb1Δ*) immunoblotted for human HSF1 using Pgk as loading control. (**g**) Summary of human HSF1 phosphorylation sites mediated by recombinant CK2α, CK2α′ or CK2 holoenzyme *in vitro* and analysed by phosphoproteomics, where (+) indicates detection of phosphorylation, DBD; DNA binding domain; LZ1-3 and LZ4; leucine zipper domains. (**h**) Yeast expressing WT human HSF1, S303A or S303/S307A mutants grown in glucose with DMSO or 10 μM TID43 and OD$_{600}$ nm monitored over 4 days. Statistical significance was measured 4 days of growth. Error bars represent ± s.e.m., (n = 3). Unpaired *t*-test; NS, no significant; *$P < 0.05$, ***$P < 0.001$.

findings were corroborated in the striatal-derived mouse HD STHdh$^{Q111}$ cell-line compared with STHdh$^{Q7}$ cells (Fig. 6a). Importantly, we tested the expression levels of HSF1, Hsp70, CK2α′ and Fbxw7 in human HD induced pluripotent stem cells (iPSC)-derived MSNs from 33Q (Control) and 60Q (HD) containing allele individuals in the absence or presence of brain-derived neurotrophic factor (BDNF; Fig. 6b). Although no differences in the expression profile of these proteins were observed

under BDNF-withdrawal stress conditions, we found that HSF1 abundance and Hsp70 expression were reduced, and CK2α′ and Fbxw7 increased, in the 60Q cell line.

**CK2α′ lowering increases HSF1 and decreases mHtt aggregation.** To explore a physiological role for CK2α′ in regulating HSF1 abundance and activity *in vivo*, the consequences of CK2α′ knock-out was evaluated in otherwise WT mice. CK2α′$^{−/−}$ mice

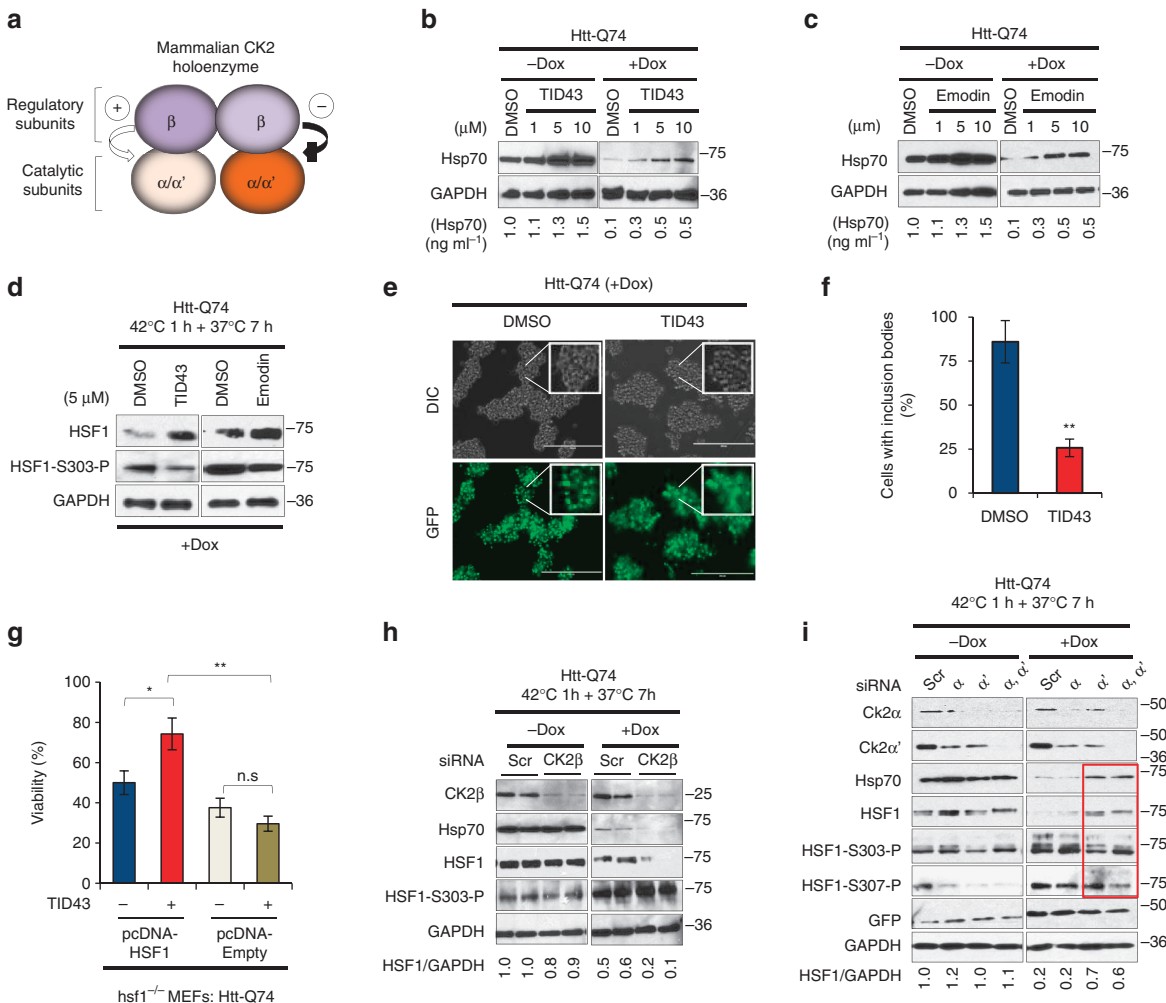

**Figure 4 | Mammalian CK2 inhibition ameliorates HSF1 degradation and mHtt aggregation and death in a cellular HD model.** (**a**) Mammalian CK2 holoenzyme subunit composition and function. (**b**) Htt-Q74 cells treated with CK2 kinase inhibitors TID43 or (**c**) Emodin 24 h before Htt-Q74 induction with Dox and heat shocked at 42 °C for 1h followed by recovery at 37 °C for 7 h and extracts analysed by immunoblotting for Hsp70 and GAPDH. (**d**) Htt-Q74 cells treated with 5 μM TID43 and immunoblotted for HSF1 and P-HSF1-S303. (**e**) Fluorescent images for GFP-Htt-Q74 analysed microscopically in cells treated with DMSO or 1 μM TID43 as described in B. Scale bar: 200 μm. (**f**) Quantification of cells containing GFP-Htt-Q74 aggregates from E expressed as percentage of total number of cells evaluated. Error bars represent ± s.e.m., ($n = 500$ cells). Unpaired $t$-test **$P < 0.05$. (**g**) hsf1$^{-/-}$ MEFs expressing Dox-inducible Htt-Q74-GFP transfected with pcDNA or WT HSF1 and incubated with 1 μM TID43 24 h before Htt-Q74-GFP induction followed by heat shock at 42 °C during 1 and 7 h recovery at 37 °C. Cell viability expressed as % of viable cells under control conditions at 37 °C. Error bars represent ± s.e.m., ($n = 3$). Unpaired $t$-test n.s., no significant, *$P < 0.05$, **$P < 0.01$. (**h**) Htt-Q74 cells were transfected with siRNA against CK2β regulatory subunit or (**i**) CK2α and/or CK2α′ catalytic subunits using scrambled siRNA (Scr) as control 24 h before Htt-Q74 induction during 2 days followed by heat shock at 42 °C 1h and recovery at 37 °C, 7 h. HSF1 was quantitated as in (F2H). All immunoblots shown for each panel contain the samples from the same membrane and were cropped to show only relevant data. See Supplementary Fig. 10 for uncropped immunoblots.

exhibited higher HSF1 protein levels and Hsp70 expression compared to CK2α′$^{+/+}$ littermates, suggesting a physiological role for CK2 in regulating HSF1 abundance and activity (Supplementary Fig. 6A). Given the dramatic elevation in CK2α′ levels in HD models and patients, mice heterozygous for CK2α′ (CK2α′$^{+/-}$) were generated in the KIQ175 background to evaluate the impact on HSF1 levels, activity and on the neuropathology observed in this HD model (Fig. 7a and Supplementary Fig. 6B). KIQ175/CK2α′$^{+/-}$ mice exhibited ∼2.5-fold decreased CK2α′, and ∼1.5-fold increased CK2α striatal mRNA levels compared to KIQ175 littermates (Fig. 7b and Supplementary Fig. 6C). Heterozygosity of CK2α′ increased expression of the HSF1 targets Hsp70 and Hsp25 (Fig. 7c and Supplementary Fig. 6D).

The co-activator peroxisome proliferator-activated receptor gamma (PGC-1α) is an HSF1 target that is a key regulator of

mitochondrial biogenesis and energy metabolism[47]. Both PGC-1α and its target genes, CYCS (cytochrome c), TFAM (mitochondrial transcription factor) and NDUFS3 (NADH dehydrogenase) are reduced in the striatum of the N171-82Q HD mouse model and in human patients, in part underlying the mitochondrial dysfunction observed in HD (ref. 48). KIQ175/CK2α′$^{+/-}$ mice showed elevated expression of PGC-1α and target genes compared to KIQ175 littermates (Fig. 7d and Supplementary Fig. 6E). Importantly, HSF1 and Hsp70 protein levels were increased, mutant Htt aggregates decreased and soluble mutant and WT Htt protein increased in the striatum of KIQ175/CK2α′$^{+/-}$ mice compared to KI175 littermates (Fig. 7e,f; Supplementary Fig. 6F,G).

**CK2α′ reduction preserves MSNs and muscle mass.** Previous studies demonstrated decreased abundance of dendritic spines

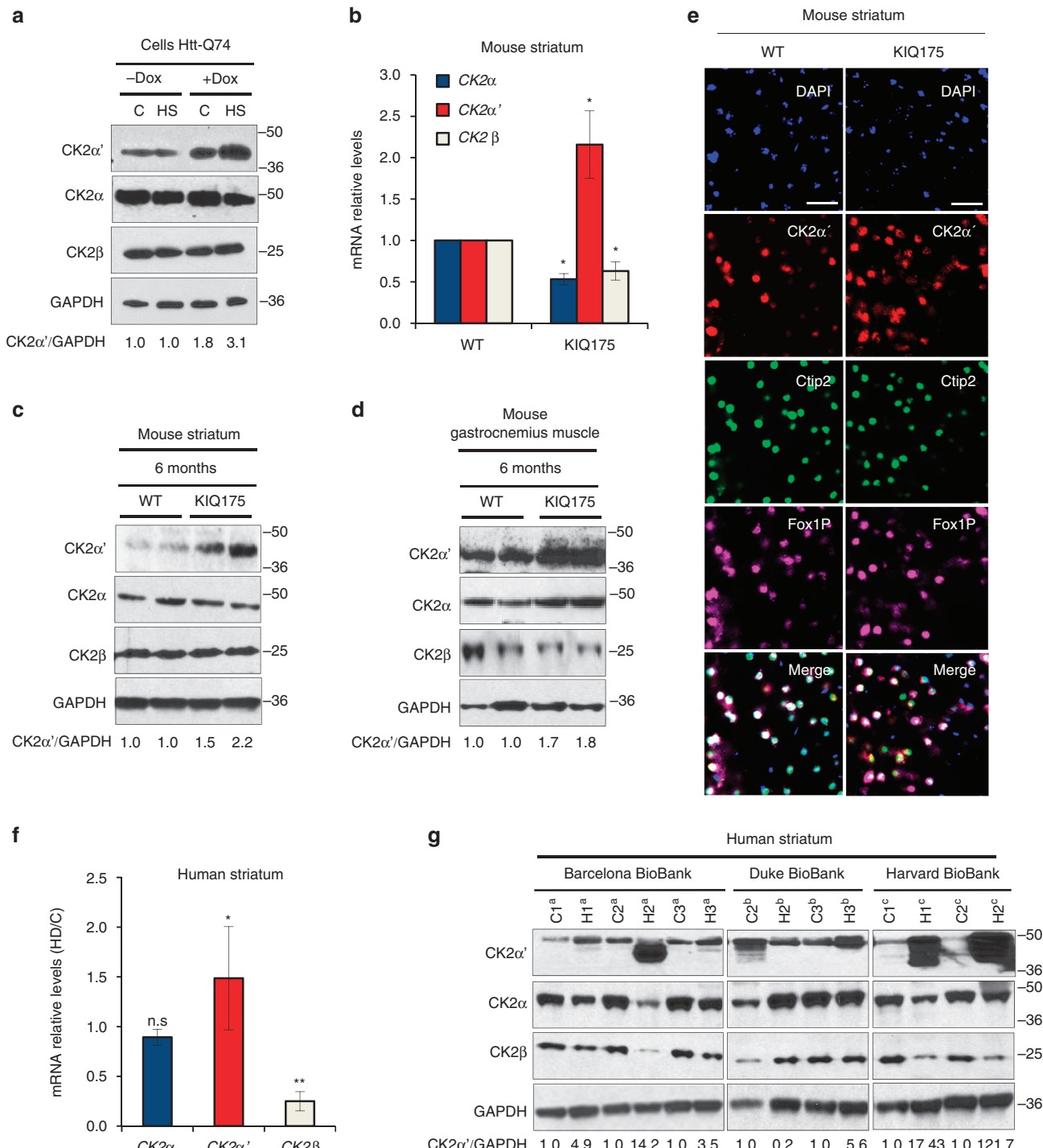

**Figure 5 | CK2α′ abundance is elevated in Huntington's disease.** (**a**) CK2α, CK2α′ and CK2β protein levels in Htt-Q74 expressing cells under control (**c**) or heat shock conditions at 42 °C for 1 h (HS). CK2 subunit abundance was quantified using Quantity One image software normalized using GAPDH as loading control. CK2α′ ratio is shown and referenced to control (−Dox) cells. (**b**) CK2α, CK2α′ and CK2β striatal mRNA levels from WT and KIQ175 mice at 6 months of age. The value given for the amount of mRNA in the control group (WT) was set as 1. Error bars represent mean ± s.e.m., (n = 4 animals). Values for the KIQ175 group were compared to the WT group. Statistical significance was measured by two-tailed unpaired t-test *P < 0.05. (**c**) Protein levels for CK2α, CK2α′ and CK2β in the striatum and (**d**) gastrocnemius muscle of WT and KIQ175 mice at 6 months of age (n = 4). (**e**) Coronal section of the striatum of WT and KIQ175 at 6 months of age, showing co-localization of CK2α′ (red) with Ctip2 (green) and Fox1p (magenta) labelled MSNs in merged image. Scale bar: 10 μm. (**f**) CK2α, CK2α′ and CK2β qRT-PCR analysis and (**g**) protein levels in the striatum of HD patients and sex-age matched controls from 3 biospecimen banks (Supplementary Table 1). The value given for the amount of mRNA in the control group (C) was set as 1 for each gene. Error bars represent mean ± s.e.m., (n = 7). One-tailed unpaired t-test *P < 0.05, **P < 0.05, NS, no significant. Values for the Huntington's disease (HD) group were compared to the control (C) group. CK2α′ bands from immunoblots were quantified using Quantity One image software (BioRad) and the protein values were normalized using GAPDH as loading control and referenced to the corresponding age-sex-matched control patient. See Supplementary Fig. 11 for uncropped immunoblots.

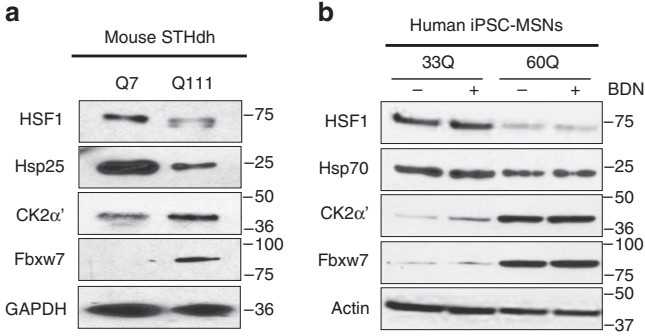

**Figure 6 | Decreased abundance of HSF1 and increased levels of CK2α′ and Fbxw7 in mouse striatal neurons and differentiated human iPSCs expressing polyQ-expanded Htt.** (**a**) Mouse STHdh Q7 and Q111 cells were cultured at 33 °C for 48 h. (**b**) Human iPSC from 33Q (Control) and 60Q (HD) individuals were differentiated into MSNs for 54 days as previously described[65,66], and transferred to minimal differentiation medium with ( + ) or without ( − ) BDNF for 48 h. Protein samples were immunoblotted for the indicated proteins. See Supplementary Fig. 12 for uncropped immunoblots.

and changes in spine morphology in HD patients and mouse HD models[49]. These phenotypes correlate with reduced expression of vital HSF1 targets in HD, such as Hsp70, which functions in dendritic maturation and neuronal communication[50]. Significant alterations in MSN morphology and composition were detected between WT and KIQ175 mice, with increased immature MSNs and decreased mature MSNs in KIQ175 mice (Fig. 8a,b). Striatal degeneration can be detected by alterations in excitatory synapse formation in KIQ175 mice at an early stage (P21) that may precede neurodegeneration[49]. Striatal inputs from the cortex and thalamus can be distinguished by the differential expression of the presynaptic proteins VGlut1 (cortico-striatal) and VGlut2 (thalamo-striatal) and their co-localization with post-synaptic PSD95 (Fig. 8c). CK2α′ heterozygosity increased thalamo-striatal excitatory synapse number (VGlut2-PSD95 co-localization) in KIQ175/CK2α′ $^{+/-}$ mice, compared with KIQ175 littermates at both 5 weeks (Supplementary Fig. 7A,B) and 6 months of age with no changes in cortico-striatal synapse number, which occurs only in older KIQ175 mice[51] (Fig. 8d–f). Since mature spines are those actively involved in synapse connectivity, the increase in mature spine number as a consequence of CK2α′ depletion (Fig. 8a,b) correlates with increased excitatory synapse number observed in this background (Fig. 8f). In addition, KIQ175/CK2α′ $^{+/-}$ mice displayed increased body weight in comparison to KIQ175 littermates and maintained similar body mass as wild type littermates beginning at 3 weeks and extending through at least 6 months (Fig. 8g,h; Supplementary Fig. 7C).

## Discussion

The primary defect in Huntington's Disease is the expression of a misfolded and aggregated polyQ-expanded Htt protein[5]. HSF1 plays a critical role in activating genes encoding proteins that function in protein quality control, mitochondrial function and cellular pro-survival factors that ameliorate defects in neuronal function and survival in HD. Activation of HSF1 either by over-expression, or by genetic or pharmacological modulation, is protective in HD models[24,28,31–34]. Previous studies using an Hsp90 inhibitor showed only short-term benefits for HSF1 activation that was attributed to global changes in chromatin structure that precluded HSF1 binding to target promoters[28]. However, our data in cellular and mouse HD models and

postmortem HD patient samples, and studies by others in flies and mice[29,31], clearly demonstrate a strong diminution in HSF1 protein abundance.

Moreover, we find that the levels of both HSF1, and HSF1 target genes, are strongly reduced in gastrocnemius muscle of the KIQ175 HD mouse. This is consistent with the accumulation of nuclear mutant Htt inclusions in muscle fibres[52], and perhaps with the cachexia observed in the periphery in this mouse model and HD patients[53]. These studies suggest an intrinsic link between the periphery and the central nervous system (CNS) in HD with respect to HSF1 abundance and activity. It has also been demonstrated that α-synuclein A53T mutant expressing cells, but not WT α-synuclein cells show HSF1 depletion; similarly, in tissue of patients with diffuse Lewy body disease HSF1 depletion is observed[54]. Rodent models of Alzheimer's disease also exhibit low HSF1 levels in the cerebellum[55], suggesting that Parkinson's disease (PD), Alzheimer's Disease (AD) and perhaps other neurodegenerative diseases associated with protein misfolding may exhibit inappropriate degradation of HSF1.

Here we demonstrate that one mechanism for defective HSF1 activation in HD is pathogenic polyQ-Htt-dependent targeting of HSF1 for proteasomal degradation, mediated by the action of CK2 kinase and the phosphorylation of HSF1 at S303/307. Our observations that CK2α′ catalytic subunit levels are dramatically elevated in MSNs in rodent HD models, in differentiated human iPSCs derived from patients with HD and in the striatum from patients with HD suggest that mutant Htt elevates the expression or stability of this kinase through as yet unidentified mechanisms. Consistent with our findings, others have shown increased expression of CK2 in striatal tissue of yeast artificial chromosome (YAC) transgenic mice expressing full-length mutant Htt (ref. 56). Moreover, AD models and patients with Alzheimer's Disease exhibit low levels of HSF1 and elevated CK2 (refs 55,57,58), and increasing HSF1 activity in an AD murine model reverses cerebellar Purkinje cell deficiency[55]. Here, we have demonstrated that CK2α′ is increased in MSNs in HD, the neuronal sub-type most susceptible to mHtt aggregation and neurodegeneration, whereas it is known that CK2α/α′ is increased in the astrocytes in the hippocampus and temporal cortex of AD patients, the most affected tissue in AD (ref. 58). Our data demonstrate that CK2α′ and mHtt are found in MSNs in HD mice, while in AD CK2 co-localizes with amyloid deposits in astrocytes. However, it is currently unclear if this causes the HSF1 protein depletion that has been reported in AD[55,58]. Notably, pharmacological inhibition of CK2 also affects the phosphorylation state of Htt at Ser13 and Ser16, which alters WT Htt protein aggregation and localization[59]. It is currently unknown what the relative contributions of these or other CK2 targets are to HD phenotype amelioration when CK2 is inhibited.

HSF1 target genes also encode proteins that function within the synaptic environment controlling dendritic maturation, presynaptic signalling pathways and neuronal communication. Moreover, Hsp70 stabilizes presynaptic proteins such as syntaxin I, synaptic vesicle protein 2 and synaptotagmin I and attenuates hypoxia-induced motor and sensory impairment[50]. HSF1 also activates BDNF expression, which contributes to the survival of peripheral and CNS neurons and that is markedly depleted in HD brain[60]. Interestingly, amyloid peptide (Aβ) transgenic mice treated with the HSF1 activator 17-AAG had increased BDNF expression in the hippocampal region and displayed attenuated Aβ-induced synaptic toxicity and memory impairment[61]. Our data suggests that reduced expression of HSF1 target genes in the striatum of HD mice and patients with HD is associated with decreased levels of HSF1. Prevention of HSF1 degradation by CK2α′ inhibition increases the expression of chaperones in the striatum. Indeed, striatal and cortical neurons, which are primary

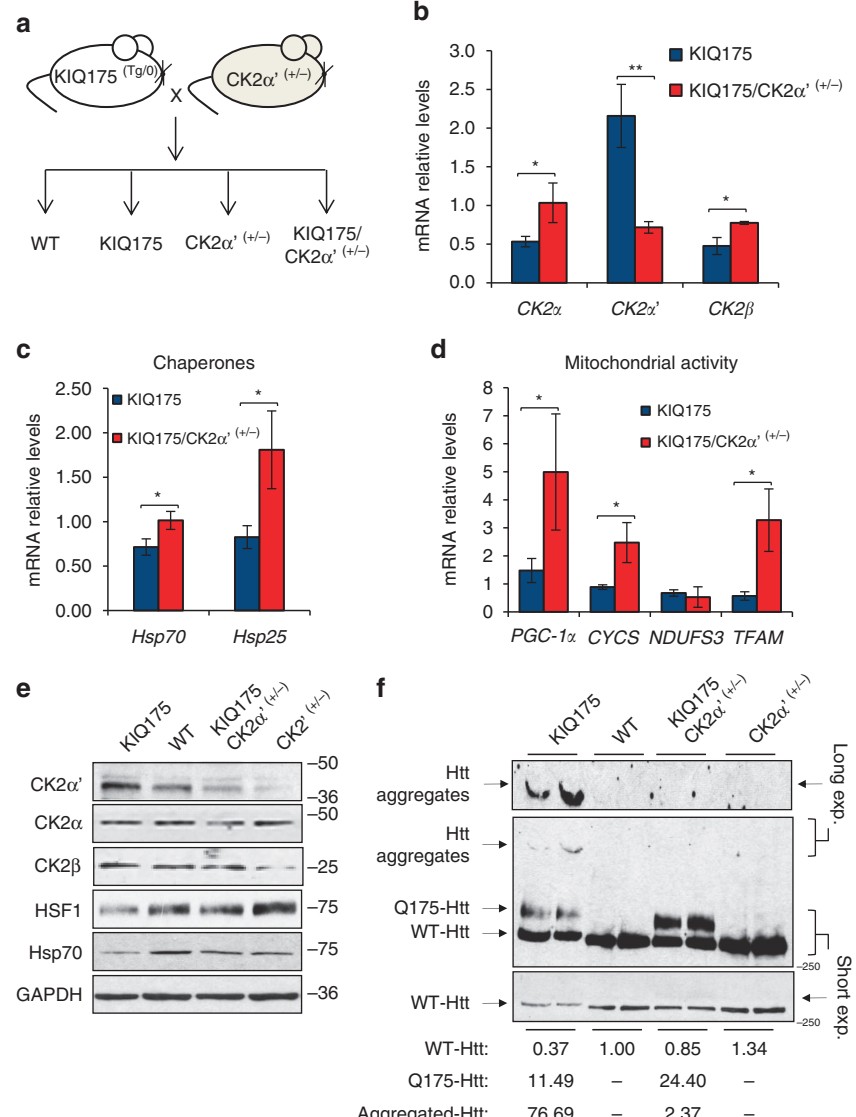

**Figure 7 | CK2α′ heterozygosity increases HSF1 abundance and activity in HD mice.** (**a**) Mouse breeding scheme to generate WT, KIQ175, CK2α′ $^{+/-}$ and KIQ175/CK2α′ $^{+/-}$ mice. (**b**) CK2α, CK2α′ and CK2β mRNA levels from KIQ175 and KIQ175/CK2α′ $^{+/-}$ striatum at 6 months. Data was normalized to GAPDH and referenced to WT. (**c**) Chaperones Hsp70 and Hsp25 and (**d**) Mitochondrial activity-related genes PGC-1α, CYCS, NDUFS3 and TFAM mRNA levels from the striatum of KIQ175 and KIQ175/CK2α′ $^{+/-}$ at 6 months. (**e**) Immunoblots of striatum from KIQ175, WT, KIQ175/CK2α′ $^{+/-}$ and CK2α′ $^{+/-}$ at 6 months. (**f**) WT and mutant Htt immunoblot analysis from striatum of KIQ175, WT, KIQ175/CK2α′ $^{+/-}$ and CK2α′ $^{+/-}$ at 6 months. Long and Short exposures are presented. mRNA data shows fold change compared to the control group (WT; Supplementary Fig. 6) which values were set as 1. Error bars represent mean ± s.e.m., ($n = 3$ animals). Unpaired t-test *$P < 0.05$, **$P < 0.05$. Values for the KIQ175/CK2α′$^{(+/-)}$ group were compared to the KIQ175 group. See Supplementary Fig. 12 for uncropped immunoblots.

sites of neurodegeneration in HD, have reduced levels of Hsp70 expression as compared with other neurons which are more resistant to mHtt aggregation[62]. Transcriptomic analysis of brain from patients with HD revealed reductions in chaperone family members and other HSF1 target genes with metabolic and synaptic functions[40] (Supplementary Data 3). Although we demonstrate that HSF1 is responsible for the decrease in chaperone expression in the striatum, other transcription factors such as p53 and NF-Y may activate Hsp70 in other areas of the brain[62,63].

The depletion of HSF1 in HD is associated with increased phosphorylation of S303 and S307 within the HSF1 central regulatory domain. This region has been previously associated with negative regulation of HSF1 through poorly understood mechanisms[21,35–37]. Nevertheless, R6/2 transgenic mice express-

ing a constitutively active form of HSF1 (lacking a 94 amino-acid of the central regulatory region encompassing S303 and S307), inhibited Htt-polyQ aggregate formation and prolonged lifespan[24]. Interestingly, in cancer cells phosphorylation of S303 and S307 by distinct protein kinases is required for the interaction of HSF1 with the Fbxw7 E3 ligase, responsible for ubiquitin-dependent degradation of nuclear HSF1 in melanoma cancer cells[41]. While cancer cells show decreased levels of Fbxw7 and elevated HSF1 protein levels, pathogenic polyQ-expressing cells and tissues show the opposite: increased Fbxw7 levels and increased CK2-dependent S303/307 phosphorylation of HSF1, which together drives HSF1 degradation (Fig. 9). The importance of maintaining appropriate levels of HSF1 may provide an avenue for therapeutic intervention for aging or diseased cells that have a decreased capacity to maintain protein homeostasis[64]. However,

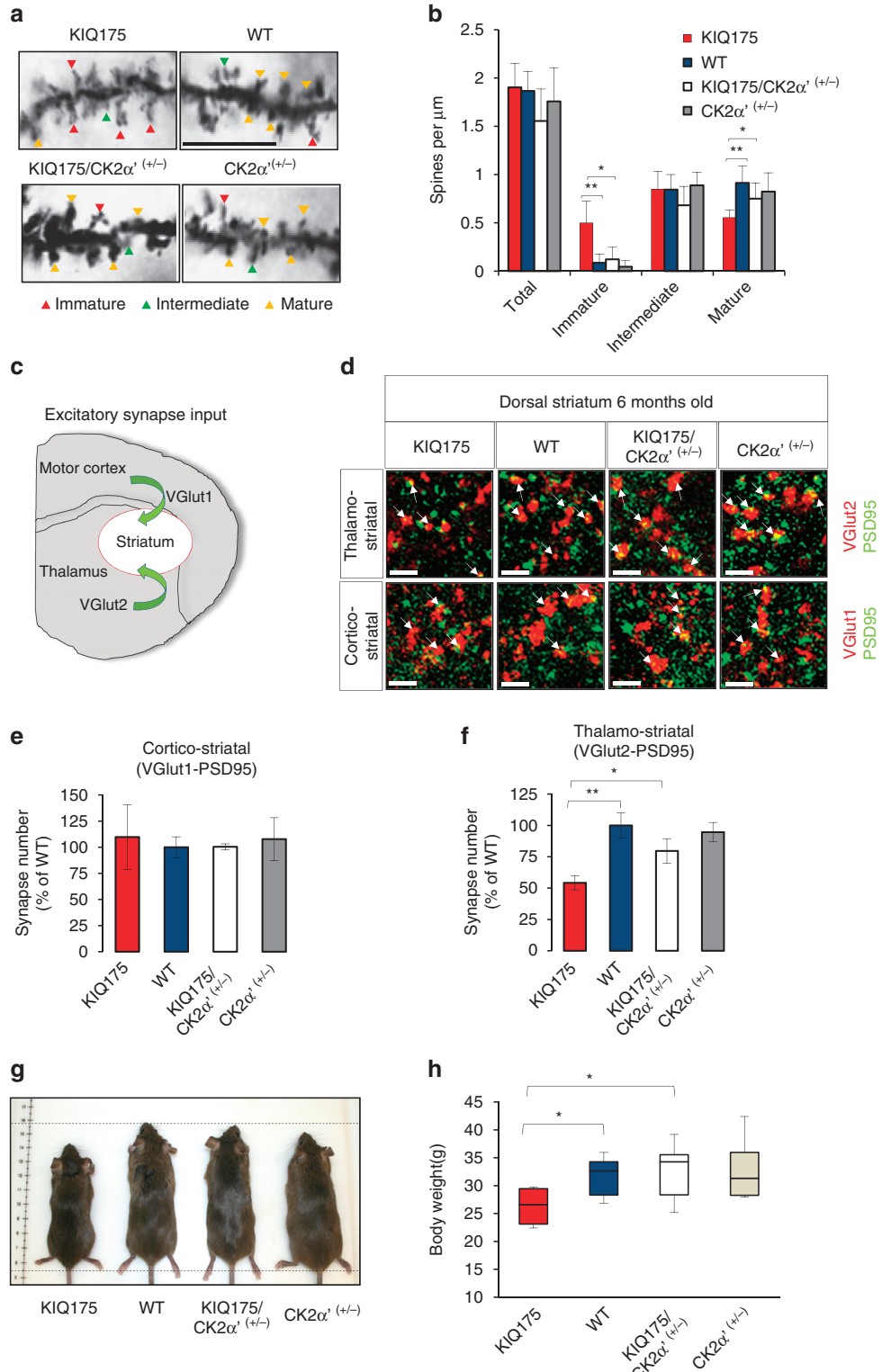

**Figure 8 | CK2α′ heterozygosity ameliorates biochemical and neurobiological defects in KI175 HD mice.** (**a**,**b**) Dendrite spine number and morphology of MSNs in the dorsal-striatum of KIQ175 and KIQ175/CK2α′$^{+/-}$ at 6 months. Scale bar, 10 μm. Error bars indicate mean ± s.e.m., ($n = 12$ cells/animal, 3 animals/genotype). Unpaired $t$-test *$P < 0.05$, **$P < 0.01$. (**c**) Excitatory synapse input in the dorsal striatum. (**d**) Immunostaining of the cortico-striatal pre-synaptic marker (VGlut1, red), the thalamo-striatal pre-synaptic marker (VGlut2, red) and the post-synaptic marker PSD95 (green) in the dorsal-striatum of KIQ175, WT, KIQ175/CK2α′$^{+/-}$, and CK2α′$^{+/-}$ at 6 months. Scale bar, 10 μM. (**e**) Quantification of VGlut1-PSD95 and (**f**) VGlut2-PSD95 co-localized synaptic puncta from B. Error bars indicate mean ± s.e.m., ($n = 3$ animals per genotype, 3 sections per animal, 15 sections per scan). Unpaired $t$-test *$P < 0.05$, **$P < 0.01$. (**g**,**h**) Size and body weight of KIQ175 ($n = 6$), WT ($n = 8$), KIQ175/CK2α′$^{+/-}$ ($n = 11$) and CK2α′$^{+/-}$ ($n = 11$) at 6 months. Error bars indicate ± s.e.m. Unpaired $t$-test *$P < 0.05$.

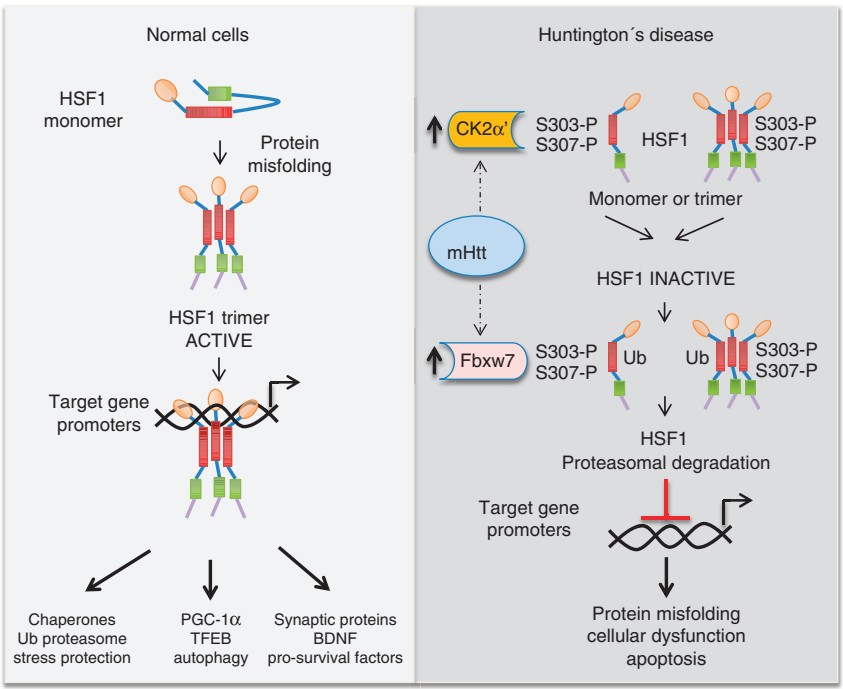

**Figure 9 | Model for HSF1 degradation in Huntington's Disease.** mHtt expression increases the abundance sof CK2α′ kinase and the Fbxw7 Fbox protein. CK2α′ phosphorylates HSF1 S303 and S307, inactivating HSF1 transcriptional activity and recruiting the Fbxw7 E3 ligase. The E3 ligase complex ubiquitinylates HSF1, targeting the protein for proteasomal degradation. Decreased HSF1 levels compromise the expression of HSF1 target genes that are essential for coping with misfolded and aggregated mHtt in Huntington's Disease and for neuronal function and survival.

further studies will be required to better understand the molecular mechanism by which poly-glutamine expanded Htt drives increased expression of the HSF1 degradation components CK2α′ and Fbxw7.

## Methods

**Cell culture and cell lines.** Mammalian cell lines used in this study were the inducible rat PC12 cells expressing HttQ23-GFP and HttQ74-GFP (a gift from Dr Rubinsztein, University of Cambridge), the inducible rat PC12 cells expressing AR-Q10 and AR-Q112 (a gift from Dr. Merry, Thomas Jefferson University), the mouse-derived striatal cells STHdh$^{Q7}$ and STHdh$^{Q111}$ (Coriell Cell Repositories), the hsf1$^{-/-}$ mouse embryonic fibroblasts (MEF) cells (from Dr. Benjamin, Medical College of Wisconsin) and hsf1$^{-/-}$ MEFs expressing Htt-Q23-GFP and Htt-Q74-GFP (prepared in this work), human HEK-293T cells (ATCC CRL-1573) and human retinal pigment epithelial cell line ARPE-19 (ATCC CRL-2302). Human HD iPSC-derived from 33Q and 60Q individuals were cultured as previously described[65] and differentiated into MSNs[66]. On day 42 of differentiation, cell aggregates were plated into 24-well plates coated with Matrigel (BD). Two weeks later cells were transferred to neural induction medium to induce MSN formation.

PC12 cells lines were maintained on Dulbecco's modified Eagle's medium (DMEM) supplemented with 5% Tet-approved foetal bovine serum (FBS), 10% horse serum, 100 μg ml$^{-1}$ G418, 75 μg ml$^{-1}$ Hygromycin B and 100 U ml$^{-1}$ penicillin/streptomycin and grown at 37 °C. MEFs were maintained in DMEM supplemented with 10% (FBS), 0.1 mM nonessential amino acids, 100 U ml$^{-1}$ penicillin/streptomycin, and 55 μM 2-mercaptoethanol and grown at 37 °C. Striatal cells were grown at 33 °C in DMEM supplemented with 10% FBS and 100 U ml$^{-1}$ penicillin/streptomycin. HEK-293T cells were maintained on DMEM supplemented with 10% (FBS) and 100 U ml$^{-1}$ penicillin/streptomycin grown at 37 °C. ARPE cells were grown in DMEM-F12 supplemented with 10% FBS and 100 U ml$^{-1}$ penicillin/streptomycin. Doxycycline induction treatments were performed using 10 μg ml$^{-1}$ for 2–3 days. All culture media were supplemented with MycoZap-Plus-PR (Lonza) for mycoplasma elimination.

PC12 cells were authenticated by immunoblotting analysis on doxycycline induction for the presence of Htt-polyQ constructs with anti-GFP antibody, AR-polyQ cells were authenticated for the presence of AR-polyQ with anti-N20 antibody. The mouse-derived striatal cells STHdh$^{Q7}$ and STHdh$^{Q111}$ (Coriell Cell Repositories) were authenticated by immunoblotting for the presence of mutant Htt with anti-Htt (Mab2166). The hsf1$^{-/-}$ MEF cells were authenticated by immunoblotting for HSF1 with anti-HSF1 (Enzo 10H8). No misidentified cells maintained by ICLAC were used in this study.

**Yeast strains and kinase inhibitor analysis.** To screen for protein kinase inhibitors that activate HSF1 in yeast we used the humanized yeast screening assay described in ref. 42. A battery of 37 different kinase inhibitors was tested. Cells were incubated in 96-well plates using a range of concentrations (2 nM-20 nM-0.2 μM-20 μM and 200 μM) for each compound using DMSO as solvent negative control and the small molecule HSF1A as positive control[18,42] and OD$_{600}$ nm was monitored over 4 days. Experiments were carried out using three independent biological replicates with three technical replicates for each experiment. Serine to Alanine mutations were introduced in pRS424-GPD-human HSF1 using site-directed mutagenesis and vectors transformed into DNY75 yeast strain. Cells were cultivated in SC-URA-TRP media containing 2% raffinose and 0.01% galactose. Overnight cultures were serially diluted and plated onto SC-URA-TRP plates containing Galactose or Dextrose. Plates were incubated for 3 days at 30 °C. TID43 treatments were carried out in flasks in SC-URA-TRP containing 4% dextrose using 10 μM TID43 or DMSO for 4 days. The *CKB1* gene was deleted in DNY75 using a loxP-KanMX4-loxP deletion cassette and transformed with pRS423-GPD (HIS) human HSF1. The *CKB1* and WT cells were grown in SC-HIS media overnight and serial dilutions plated onto SC-HIS and 5-FOA plates and incubated for 3 days at 30 °C. Protein extracts were prepared from cells after overnight growth for HSF1 protein analysis.

**Generation of hsf1$^{-/-}$ MEF cells expressing Htt-polyQ-GFP.** The hsf1$^{-/-}$ MEFs expressing polyQ-Htt-GFP were generated by amplification of the Htt exon 1 Q23-GFP and Q74-GFP DNA fragments from PC12 cells by using F1-AscI-GFP primer 5′-ATATGGCGCGCCATGGTGAGCAAGGGCGAGGAGCTGTTCACCG GGGTGG-3′ and R1-MluI-Exon1Htt 5′-ATATACGCGTTCACGGCGGGGGCG GCGGCGGGGGCGGC-3′. HttQ23-GFP and HttQ74-GFP DNA fragments were cloned into pTRIPz-TeT-ON plasmid (derived from original pTRIPz empty vector from Thermo Scientific). pTRIPz empty vector was digested with AgeI and MluI removing tRFP from the vector and introduced a new multicloning site (MCS) 5′-ACCGGTGAATTCGGCGCGCCATCGATCCTGCAGGCTCGAGACGCGT-3′ containing an AscI site. AscI and MluI restriction enzyme sites were used to subclone the Htt-Q23-GFP and Htt-Q74-GFP encoding DNA fragments into pTRIPZ-TeT-ON vector. HEK-293T cells were transfected to produce lentiviral particles using pTRIPZ-TeT-GFP-HttQ23 or pTRIPZ-TeT-GFP-HttQ74, psPAX2 (packaging vector) and VsgV (envelope vector). Lentiviral particles were used to infect hsf1$^{-/-}$ MEFs and positive clones were selected in 2.5 μg ml$^{-1}$ Puromycin.

**Cell transfection and siRNA knockdown experiments.** Cells were transfected using Lipofectamine LTX Reagent (Invitrogen) and following Invitrogen protocol instructions (Invitrogen protocols 25-0946W) or SE Cell Line Nucleofector Kit. HttQ74 cells were transfected with HA-Ub vector (Addgene #18712) by using

Lipofectamine LTX Reagent (Invitrogen) following protocol instructions (Invitrogen protocols 25-0946 W). hsf1$^{-/-}$ MEF-GFP-Htt-Q74 cells were transfected with pcDNA WT HSF1 or the HSF1 S303A mutant plasmid using Lipofectamine LTX reagent. hsf1$^{-/-}$ MEFs were electroporated with FBXW7-FLAG tag and WT HSF1 or S303A plasmids using the SE Cell Line Nucleofector.

Knockdown experiments were carried out using FlexiTube siRNA (5 nmol) from Qiagen for the knockdown of HSF1 and CK2 subunits CK2α, CK2α′, CK2β, the GSK3 subunits, GSK3α, GSK3β and HttQ74 cells were used with FlexiTube siRNA (5 nmol) from Qiagen; Rn_Hsf1_4 SI01523144, Rn_Csnk2α1_4 SI02007180, Rn_Csnk2α2_4 SI04730642, Rn_Csnk2β_4 SI01994384, Rn_Gsk3α_4 SI01997016, Rn_Gsk3β_4 SI01519420. For FBXW7 knockdown the mouse-derived STHdhQ7 and STHdhQ111 cells were used with a combination of 4 siRNAs (Mm_Fbxw7; SI05586490; SI05586483; SI01001056; SI01001049). As negative control for silencing experiments, an ON-TARGETplus non-targeting siRNA (Scrambled) was obtained from Dharmacon. Knockdown experiments in PC12 Htt-Q74 cells were carried out using DharmaFECT1 transfection reagent 24 h before 10 μg ml$^{-1}$ doxycycline induction for 2 days. Knockdown experiments in mouse-derived striatal cells were carried out using DharmaFECT1 transfection and cells were harvested 48 h after transfection.

**Viability assays.** Cell viability was assessed using the XTT Cell viability assay kit (Roche) following manufacturer's recommendations.

**RNA preparation and RT-PCR.** RNA was extracted from PC12 HttQ74 cells and human tissue by using RNEasy extraction kit (Qiagen) according to manufacturer's instructions. For worm RNA extraction, 50 worms were incubated in a thermo-cycler at 65 °C for 10 min and RNA extraction was performed using phenol/chloroform, precipitated with one volume of 100% isopropanol and 20 μg glycogen and resuspended in water. cDNA for all samples was prepared from 1 μg RNA using the Supercript First Strand Synthesis System for RT-PCR kit (Invitrogen) according to manufacturer's instructions. SYBR green based PCR was performed with SYBR mix (BioRad).

**Chromatin immunoprecipitation.** For chromatin immunoprecipitation assays Htt-Q74 cells were grown in 10 cm plates to 75% confluency, HttQ74 expression was induced with 10 μg ml$^{-1}$ doxycycline for 3 days and heat shocked at 42 °C for 30 min. Plates were immediately placed on ice and cross-linked with 500 μl Formadehyde for 5 min at 4 °C. The reaction was terminated by the addition of Glycine to a final concentration of 125 mM and incubated on ice for 5 min. Cells were transferred to a 15 ml falcon tube and centrifuged at 3,000 r.p.m. for 5 min. Cells were lysed by adding 1 ml cell lysis buffer (25 mM Tris pH 7.5, 150 mM NaCl, 1 mM EDTA, 1% Triton X-100, 0.1% SDS) and incubated on ice for 10 min. Lysates were equilibrated in 1 ml immunoprecipitation buffer (IPB) (50 mM Tris pH 7.5, 150 mM NaCl, 1 mM EDTA, 1% Triton X-100) and sonicated three times for 30 s. Samples were centrifuged at 12,000 r.p.m. during 10 min. An aliquot (30 μl) was saved for control input DNA. 5 μg affinity purified anti-HSF1 antibody (Bethyl)[42] was added and samples were incubated overnight at 4 °C with constant rocking. Protein G agarose beads were added and incubated for 4 h at 4 °C. Immunocomplexes were washed, eluted using elution buffer (10 mM Tris-HCl, pH 8, 1 mM EDTA, pH 8, 1% SDS), and crosslinking was reversed at 65 °C for 12 h. Protein was digested by addition of 5 μl Proteinase K and incubated at 37 °C for 2 h. Chromatin was purified using the Qiaquick min-elute PCR purification kit (Qiagen) per the manufacturer's instructions. SYBR green quantitative PCR was carried out on pulldown and input samples and the ΔCt method was used to determine relative amounts of DNA. Binding of HSF1 was evaluated using primers spanning the heat shock element of the Hsp70 gene promoter 5′-TGACCTTTCCTGTCCATTCC-3′ and 5′-CAGATCTGGGGGTTAGCTGGA-3′ and values were normalized against binding to the actin promoter.

**Immunoprecipitation assays.** For HSF1 immunoprecipitation cell lysates were prepared using 500 μl of 1× cell lysis buffer (20 mM HEPES, 5 mM MgCl2, 1 mM EDTA, 100 mM KCl, 0.03% NP-40), incubated on ice 10 min and centrifuged at 14,000 r.p.m. for 10 min at 4 °C. Protein was quantified by the bicinchoninic acid assay (BCA) method (Pierce) and all samples were diluted to 1 μg ml$^{-1}$. An aliquot of the cell lysate (25 μl) was mixed with 6× SDS, boiled at 95 °C for 5 min and subjected to immunoblot analysis to validate protein levels used for immunoprecipitation (referred as whole-cell extract). Samples were pre-cleared with 50 μl Protein G Dynabeads (Life Technologies) to remove nonspecific interactions for 5 h at 4 °C. Beads were removed using a magnet DYNAL (Invitrogen). Samples were transferred to a new tube and incubated overnight at 4 °C in the presence of 5 ug anti-HSF1 antibody (Bethyl)[42] with constant rocking. 50 μl Protein G Dynabeads (Life Technologies) were added to the samples and incubated for 6 h at 4 °C with constant rocking. Beads were washed with 0.1 M Na-citrate pH 5.3 during 3 times and pull-down antibody was eluted in 50 μl 0.1 M Na-citrate pH 2. Sample was neutralized with 10 μl 1.5 M Tris-HCl pH 8.8. Sample was mixed with 6× SDS, boiled at 95 °C for 5 min and subjected to immunoblotting. For HA-Ub immunoprecipitation cells lysates were prepared in 500 μl of 1× cell lysis buffer (5% glycerol, 25 mM Tris-HCl pH 7.4, 150 mM NaCl, 1 mM EDTA, 1% NP-40), incubated on ice during 10 min and centrifuged at 14,000 r.p.m. for 10 min at 4 °C.

Anti-HA magnetic beads (Thermo Scientific) were incubated with protein samples for 1 h at room temperature. Elution was carried out in 0.1 M Glycine pH 2.0 and samples were neutralized using 1M Tris pH 8.5. For FBXW7-FLAG tag immunoprecipitation cell lysates were prepared in 500 μl of 1× cell lysis buffer (50 mM Tris pH 7.4, 10% glycerol, 150 mM NaCl, 1 mM EDTA, 1% triton X-100), incubated on ice during 10 min and centrifuged at 14,000 rpm for 10 min at 4 °C. Anti-FLAG M2 magnetic beads (Sigma) were incubated with protein samples overnight at 4 °C. Elution was carried out in 1× TE, 10% SDS at 65 °C for 10 min.

**Mouse strains.** For this study we used a full-length knock-in mouse model of HD known as zQ175, which harbours a chimeric human/mouse exon 1 carrying an expansion of ∼175 CAG repeats and the human poly-proline region[38], at 2, 6 and 12 months of age comparing to WT (C57BL/6) animals at the same age. Two females and two males were use for each genotype and age. Breeding and genotyping conditions are detailed in Supplemental Experimental Procedures. For analysis on the effect of the absence of CK2α′ in the HSF1 protein abundance we obtained CK2α′ heterozygous mice (CK2α′$^{(+/-)}$) males and females from Dr Seldin. For our analyses on the effect of the absence of CK2α′ in the KIQ175 background (C57BL/6) we crossed males CK2α′$^{(+/-)}$ and females Htt$^{(zQ175/+)}$ generating the four genotypes; WT (CK2α′$^{(+/+)}$ Htt$^{(+/+)}$), CK2α′$^{(+/-)}$ (CK2α′$^{(+/-)}$ Htt$^{(+/+)}$), KIQ175 (CK2α′$^{(+/+)}$ Htt$^{(zQ175/+)}$), KIQ175/CK2α′$^{(+/-)}$ (CK2α′$^{(+/-)}$ Htt$^{(zQ175/+)}$). Animals were analysed at 5 weeks and 6 months of age. Sample size was set to a minimum of three animals per genotype for every analysis. For body weight monitoring only males were evaluated. No randomization of animals was used in this study. The IACUC of Center of Animal Care and Use at Duke University approved the animal protocol used in this project (approval number: A173-14-07).

**Immunohistochemistry.** Mice were perfused intra-cardially with tris-buffered saline (TBS) (25 mM Tris-base, 135 mM Nacl, 3 mM KCl, pH 7.6) supplemented with 7.5 μM heparin followed with 4% PFA in TBS as previously described in ref. 49. Brains were dissected, fixed with 4% PFA in TBS at 4 °C overnight, cryoprotected with 30% sucrose in TBS overnight and embedded in a 2:1 mixture of 30% sucrose in TBS:OCT (Tissue-Tek). Brains were cryo-sectioned at 20 μm using a Leica CM3050S, washed and permeabilized in TBS with 0.2% Triton X-100 (TBST). Sections were blocked in 5% normal goat serum (NGS) in TBST for 1 h at room temperature. Primary antibodies were diluted in 5% NGS in TBST: mouse anti-1C2 (mHtt) 1:500 (Millipore), rat anti-HSF1 1:500 (Enzo SPA-950-F), rabbit anti-HSF1-S303-P 1:1,000 (Ab47369), rabbit anti- CK2α′ (EAP0505) (Elabscience) 1:200, rat anti-Darpp32 (R&D Systems) 1:1,000, mouse anti-Glutamine Synthetase (BD Biosciences 610517) 1:1,000, rat anti-CD68 (clone FA-11) (BioLegend 137002) 1:1,000, mouse anti-NeuN (clone A60) (Millipore MAB377) 1:1,000, mouse anti-Fox1P (JC12) (Abcam) 1:500, rat anti-Ctip2 (Abcam) 1:500. Sections were incubated overnight at 4 °C with primary antibodies. Secondary Alexa-fluorophore-conjugated antibodies (Invitrogen) were added (1:200 in TBST with 5% NGS) for 2 h at room temperature. Slides were mounted in Vectashield with DAPI (Vector laboratories), and images acquired on a confocal laser-scanning microscope (Leica SP5).

**Synapse quantification in mouse brain sections.** Three independent coronal brain sections were used for each mouse, containing the dorsal striatum (bregma 0.5–1.1 mm) and were stained with presynaptic VGlut1 or VGlut2 (Chemicon, anti-guinea pig, 1:500) and postsynaptic PSD95 (Zymed, Rabbit, 1:500) markers as described previously[49]. Secondary antibodies used were goat anti-guinea pig Alexa 488 (VGlut1/2) dilution 1:200 and goat anti-rabbit Alex 594 (PSD95) dilution 1:200 (Invitrogen). Three mice for each genotype; WT, CK2α′$^{(+/-)}$, KIQ175 and KIQ175/CK2α′$^{(+/-)}$ were evaluated in a double-blinded fashion. The 5 μm-thick confocal scans (optical section depth 0.33 μm, 15 sections per scan, imaged area per scan = 20.945 μm$^2$) of the synaptic zone in dorsal striatum were performed at 63× magnification on a Leica SP5 confocal laser-scanning microscope. Maximum projections of three consecutive optical sections (corresponding to 1 μm depth) were generated. The Puncta Analyzer Plugin for ImageJ (available upon request; c.eroglu@cellbio.duke.edu) was used to enumerate co-localized synaptic puncta. This assay takes the advantage of the fact that presynaptic and postsynaptic proteins reside in separate cell compartments (axons and dendrites, respectively), and they would appear to co-localize at synapses because of their close proximity. The number of animals used in our study was 12 mice for each genotype group and sex. At least 5 optical sections per brain section and at least 3 brains sections per animal were analysed, making a total of 45–60 image data sets per brain region in each genotype/age.

**Golgi Cox staining and dendritic spine analysis.** Golgi Cox staining was performed on WT, CK2α′$^{(+/-)}$, KIQ175 and KIQ175/CK2α′$^{(+/-)}$ mice (three mice per genotype) using FD Rapid GolgiStain Kit (FD NeuroTechnologies). Dye-impregnated brains were embedded in Tissue Freezing Medium (TFM, TBS), rapidly frozen on ethanol pretreated with dry ice, cryo-sectioned coronally at 200 μm thickness and mounted on gelatin-coated microscope slides (Southern Biotech). Sections were stained according to directions provided by the manufacturer. Sections that contain the dorsal striatum were imaged and MSNs in

the striatum were identified by their morphology. Secondary and tertiary apical dendrites were imaged for spine analysis as follows: z-stacks (30 μm total on z-axis, single section thickness = 0.5 μm) of Golgi-stained dendrites were taken at 63 × magnification on a Zeiss AxioImager M1 microscope. A series of TIFF files corresponding to each image stack were loaded into the Reconstruct programme (http://synapses.clm.utexas.edu) and spine analyses performed as previously described[49,67]. The classification of spines is based on width, length, and length:width ratio measurements taken using the Reconstruct software (http:// synapses.clm.utexas.edu; RRID:nif-0000-23420) designed and validated[67]. Spines were identified and classified by choosing 10 μm segments of dendrites and identified on selected dendritic stretches. The z-length (spine length) and spine head width were measured for each spine. Measurements were exported to a custom Microsoft Excel macro that was used to classify spines based on the width, length, and length:width ratio measurements taken in Reconstruct. Spines were categorized based on the following hierarchical criteria: (1) more than one spine head = 'branched spine,' (2) head width > 0.7 μm = 'mushroom spine,' (3) length > 2 μm = 'filopodia,' (4) length:width > 1 = 'thin spine,' and (5) length:width ≤ 1 = 'stubby spine.' Branched and mushroom spines were identified as mature spines, thin and stubby spines were categorized as intermediate spines, and filopodia were classified as immature spines. Statistical analyses of changes in spine density, length, width and spine type were conducted in the Statistica programme (StatSoft): A total of 3 animals per genotype, 15 dendrites per animal, 45 dendrites per genotype were analysed in MSNs in the dorsal striatum in a blinded fashion. The number of spines analysed per neuron type per age per genotype exceeded 1,000.

**Human samples.** HD Brain tissues were obtained from three independent sources; from Banc de Teixits Neurologics Biobanc Hospital Clinic-IDIBAPS (Barcelona Brain Bank) via Dr Isidre Ferrer[68] following the guidelines of the local ethics committees; from Duke Kathleen Price Bryan Brain Bank and from Harvard Brain Tissue Resource Center. Cases with and without clinical neurological disease were processed in the same way following the same sampling protocols. Control and HD cases were compared pairwise for sex, age and postmortem time (Supplementary Table 1). Brain tissue was homogenized in 25 mM Tris-HCl pH 7.4, 150 mM NaCl, 1 mM EDTA, 0.1% SDS, 1% Triton X-100. Extra SDS was added to the suspension to a final concentration of 2% (w per v) and heated at 95 °C for 5 min. The homogenate was sonicated 3 times 45 s on ice, centrifuged at 12,000 r.p.m. for 10 min and protein concentration determined using the BCA protein quantification assay (Pierce). A total of 60 μg protein was loaded onto SDS–PAGE for immunoblotting.

**mHtt expression and aggregation analysis.** Protein samples from mouse striatum at 6 months were prepared in cell lysis buffer (25 mM Tris pH 7.4, 150 mM NaCl, 1 mM EDTA, 1% Triton-X100 and 0.1% SDS). Extra SDS was added to the suspension to a final concentration of 2% (w per v) and lysates heated at 95 °C for 5 min to solubilize tissue. Total tissue homogenate was then sonicated three times 45 s on ice and centrifuged at 12,000 r.p.m. for 10 min. Samples were subjected to immunoblotting using a 7.5% stain-free criterion gel and transferred into PDVF membrane under semi-dry conditions for 1 h at 15 V. WT and mutant Htt expression and Htt aggregates were visualized using the Anti-Htt antibody Mab2166 (Millipore). Analysis of Htt-Q74 aggregation in the presence of TID43 kinase inhibitor in PC12 cells was assessed by fluorescence microscopy. Htt-Q74-GFP expressing cells were seeded into a 6-well plate and treated with either DMSO or 1 μM TID43 for 6 h before 10 μg ml$^{-1}$ doxycycline followed by a 48 h incubation. Cells were heat shocked for 1 h at 42 °C. Fluorescence was analysed using a Zeiss Axio Observer Fluorescence microscope. For quantification of fluorescence microscopy, ~500 cells were counted for each treatment. The number of cells containing aggregates was calculated as a percentage of the total number of cell counted. Analysis of Htt aggregation in the Htt-Q74 cell line was analysed by immunoblotting. Cells were seeded (5 × 10$^5$ cells per well) into a 6-well plate and treated with DMSO, SB21 or TID43 (1 μM) for 6 h before 10 μg ml$^{-1}$ doxycycline induction for 3 days. Cells were heat shocked for 1 h at 42 °C followed by 6 h recovery at 37 °C. Extracts were prepared in cell lysis buffer (25 mM Tris pH 7.4, 150 mM NaCl, 1 mM EDTA, 1% Triton-X100 and 0.1% SDS), and soluble and insoluble fractions were separated by centrifugation and analysed by immunoblotting using GFP antibody as marker for Htt-Q74-GFP presence in the pellet fraction. A filter retardation assay was carried out on striatum samples from 6 month old WT, KIQ175, CK2α′$^{+/-}$ and KIQ175/CK2α′$^{+/-}$ mice. Total tissue protein extract (175 μg protein) was loaded onto a cellulose acetate membrane and probed for Htt aggregation with Mab2166 (Millipore) and in parallel samples were loaded onto nitrocellulose membrane to probe for GAPDH as loading control.

**Protein expression and purification.** A DNA cassette encoding a codon-optimized human HSF1 was sub-cloned into the pET15b expression vector containing an amino-terminal His$_6$ tag using NdeI and XhoI to generate hHSF1-pET15b. The resulting plasmid was transformed into *Escherichia coli* strain BL21(DE3). Over-night cultures were diluted 1:100 and grown to OD$_{600}$ = 0.6 at 37 °C. Cultures were transferred to 15 °C, induced with 1 mM isopropyl 1-thio-β-D-galactopyranoside, and grown for 16 h. Cell pellets were lysed in nickel-nitrilotriacetic acid buffer (NB:

50 mM HEPES, pH 7.5, 300 mM NaCl) supplemented with 20 mM imidazole HCl (Im-HCl), using sonication three times with 30-s bursts. Lysates were cleared by centrifugation at 20,000 g for 30 min and incubated with 2 ml (bed volume) of nickel-nitrilotriacetic acid-agarose beads (Qiagen) per liter of culture. Beads were washed twice with NB + 40 mM Im-HCl, twice with NB supplemented with 40 mM Im-HCl, 5 mM ATP, and 20 mM MgCl$_2$, and once with NB + 40 mM Im-HCl. Bound protein was eluted with NB + 250 mM Im-HCl. Eluted proteins were separated on a Sephacryl S400 (GE Healthcare) gel filtration column using an ÄKTA FPLC (GE Healthcare) at a flow rate of 1.3 ml per min in 25 mM HEPES, pH 7.5, and 150 mM NaCl at 4 °C. Fractions corresponding to HSF1 monomer and HSF1 trimer were collected, pooled, concentrated and aliquoted at 10 μM (~0.6 mg ml$^{-1}$), flash-frozen in N$_2$, and stored at − 80 °C. GST-CK2α and GST-CK2α′ expression vectors were purchased from Addgene (pDB1 #27083 and pDB6 #27084, respectively). Vectors were transformed into *E. coli* strain BL21 (DE3). Overnight cultures were diluted 1:100, grown to OD$_{600}$ = 0.6 at 37 °C and induced with 1 mM isopropyl 1-thio-β-D-galactopyranoside for 5 h at 37 °C. Cell pellets were lysed in GST equilibration buffer (50 mM Tris, 150 mM sodium chloride, pH 8.0) supplemented, using sonication three times with 30 s bursts. Lysates were cleared by centrifugation at 20,000g for 30 min and incubated with 2 ml (bed volume) of Glutathione agarose beads (Pierce) per liter of culture. Beads were washed twice with equilibration buffer and GST-CK2 subunits eluted with equilibration buffer (50 mM Tris, 150 mM sodium chloride, pH 8.0) supplemented with 10 mM reduced Glutathione. Eluted proteins were concentrated using 5,000 MWCO centricon, aliquoted (~5 mg ml$^{-1}$), flash-frozen in N$_2$ and stored at − 80 °C.

*In vitro* **phosphorylation of HSF1 by CK2.** Quantification of the specific activity (U per ml) of GST-CK2α (Addgene pDB1-27083) and GST-CK2α′ (Addgene pDB6-27084) purified recombinant enzymes was performed using casein kinase assay kit (CycLex) and commercial CK2 holoenzyme (New England Biolabs) as positive control for phosphorylation. Quantification was carried out following per manufacturés instructions. Purified recombinant human HSF1 trimer (1 μg) was incubated with recombinant CK2 (Holoenzyme or GST purified subunits CK2α or CK2α′) with 5,000 U ml$^{-1}$. As negative controls for the assay we used samples without ATP or samples without enzyme. The reaction was carried out at 37 °C during 30 min (control conditions) or at 37 °C during 20 min followed by 10 min at 42 °C (Heat shock conditions). 6 × SDS was added and boiled for 2 min to terminate the reaction. Samples were analysed by phosphoproteomics.

**Immunoblot analysis.** Protein samples were separated on 4–20% SDS Criterion TGX Stain-Free gels (BioRad) at 110 V. Proteins were transferred to a nitrocellulose membrane (BioRad 0.2 μm) in Tris–Glycine Buffer (25 nM Tris-Base, 200 mM Glycine) at 25 V for 30 min in Trans Turbo Transfer system (BioRad). The membrane was blocked with 5% non-fat dry milk in TBS containing 0.25% Tween-20 (TBST) for 1 h at room temperature, incubated with primary antibodies 1:1,000 in TBST containing 2.5% milk overnight at 4 °C, and then washed 3 times for 15 min each in TBST followed by incubation with secondary antibodies Amersham ECL HRP Conjugated Antibodies 1:5,000 in TBST containing 2.5% milk (GE Healthcare) for 1 h at room temperature. After washing three times for 15 min each in TBST bands were detected with SuperSignal Chemiluminescent substrate (Thermo Scientific). Primary antibodies used in this study were used at 1:1,000 and are: anti-FLAG (M2, Sigma), anti-HA (Y-11, Santa Cruz), anti-Pgk1 (22C5, Invitrogen), anti-GAPDH (6C5, Santa Cruz), anti-GFP (Sc8334, Santa Cruz), rabbit anti-HSF1 (Bethyl[42]), rat anti-HSF1 (10H8, Enzo), anti-HSF1-S303P (Ab47369, Abcam), anti-HSF1-S307 (sc135640, Santa Cruz), anti-Hsp70 (C92F3A-5, Enzo), anti-Hsp25 (ADI-SPA-801-F, Enzo), anti-CK2α (Ab6040, Abcam), anti-CK2α′ (Ab75309, Abcam), anti-CK2β (Ab6025, Abcam), anti-FBXW7 (Ab109617, Abcam), anti-Htt (Mab2166, Chemicon) and (Mab1574, Millipore). Anti-Hsp60 (SPA-807, Stressgene), anti-Hsp90 (ADI-SPA-846F, Enzo), anti-Erdj3 (C-7, Santa Cruz), anti-Hspb5 αβ-Crystallin (F-10, Santa Cruz), anti-Bag3 (ALX-803-323-C100, Enzo Life Sciences), anti-AR (N-20, sc816, Santa Cruz), anti-panGsk3 (IH8, Santa Cruz).

**C. elegans experiments.** All *Caenorhabditis elegans* strains were maintained on normal growth media seeded with *E. coli* OP50 until required for experiments. To perform RNAi against *kin-3*, *E. coli* HT115 expressing *kin-3* double-stranded RNA or the vector L4440 (control) was cultured in LB broth containing ampicillin (50 μg ml$^{-1}$) for 8 h. Cultures were concentrated (10 ×) and seeded onto normal growth media plates containing isopropyl-1-thio-β-D-galactopyranoside (3 mM) and ampicillin (50 μg ml$^{-1}$). Plates were incubated overnight at 37 °C and allowed to cool before plating animals. For each strain, Q37::YFP (rmIs225[Punc-54::q37::yfp]II) and Q37::YFP;hsf-1(sy441) (from Dr. Morimoto, Northwestern University), L4s were plated on RNAi plates and grown for a total of 5 days at 20 °C. To prevent progeny from contaminating plates, animals were transferred every two days.

**Phosphoproteomic analysis.** Phosphoproteomic analysis was performed by the Proteomics and Metabolomics Shared Resource at Duke University. Protein samples were mixed with loading buffer and reduced with 10 mM DTT at 70 °C for 10 min before SDS–PAGE separation on a 4–12% bis-tris acrylamide gel

(NuPAGE, Invitrogen) with colloidal coomassie staining. Bands corresponding to HSF1 were excised and subjected to standardized in-gel trypsin digestion (http://www.genome.duke.edu/cores/proteomics/sample-preparation/documents/In-gelDigestionProtocolrevised.pdf). Extracted peptides were lyophilized to dryness and resuspended in 12 μl of 0.2% formic acid/2% acetonitrile. Phosphopeptides were enriched using GL Biosciences p10 TiO2-derivatized tips according to the manufacturer's protocol. Each sample was subjected to chromatographic separation on a Waters NanoAquity UPLC equipped with a 1.7 μm BEH130 $C_{18}$ 75 μm I.D. X 250 mm reversed-phase column. The mobile phase consisted of (A) 0.1% formic acid in water and (B) 0.1% formic acid in acetonitrile. Following a 4 μl injection, peptides were trapped for 3 min on a 5 μm Symmetry $C_{18}$ 180 μm I.D. X 20 mm column at 5 μl per min in 99.9% A. The analytical column was then switched in-line and a linear elution gradient of 5% B to 40% B was performed over 30 min at 400 nl per min. The analytical column was connected to a fused silica PicoTip emitter (New Objective, Cambridge, MA) with a 10 μm tip orifice and coupled to a QExactive Plus mass spectrometer through an electrospray interface operating in a data-dependent mode of acquisition. The instrument was set to acquire a precursor MS scan from m/z 375–1675 with MS/MS spectra acquired for the ten most abundant precursor ions. For all experiments higher-energy collisional dissociation (HCD) energy settings were 27v and a 120 s dynamic exclusion was employed for previously fragmented precursor ions.

Raw LC-MS/MS data files were processed in Proteome Discoverer (Thermo Scientific) and then submitted to independent Mascot searches (Matrix Science) against a SwissProt database (Human taxonomy) containing both forward and reverse entries of each protein (20,322 forward entries). Search tolerances were 5 p.p.m. for precursor ions and 0.02 Da for product ions using trypsin specificity with up to two missed cleavages. Carbamidomethylation (+57.0214 Da on C) was set as a fixed modification, whereas oxidation (+15.9949 Da on M), deamidation (+0.98 Da on NQ), and phosphorylation (+79.99 Da on STY) were considered dynamic mass modifications. All searched spectra were imported into Scaffold (v4.3, Proteome Software) and scoring thresholds were set to achieve a peptide false discovery rate of 1% using the PeptideProphet algorithm. Raw data can be found in Supplementary Data 1 and 2.

**Statistical analysis.** The t-test was used to compare mean values in groups of samples for all experiments. Error bars were calculated using mean ± s.e.m. with a group size n ≥ 3. All reported P-values were calculated for groups with unequal variance using the Excel software programme (Microsoft) and a two-tailed unpaired t-test. Statistical significance of CK2 subunits mRNA expression between control groups and HD patients group (n = 7) was analysed by a one-tailed unpaired t-test. Reported P-values for all experiments correspond to *$P < 0.05$, **$P < 0.01$, ***$P < 0.001$; NS, not significant.

**Data availability.** All data generated or analysed during this study are included in this published article (and its Supplementary Information files).

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

## Acknowledgements

We thank Drs D. Rubinsztein and D. Merry for cell lines, Glyn Noguchi, Rhona Ke and Caley Burrus for technical assistance, Dr Isidre Ferrer for providing the Barcelona Biobank human samples and Dr M. Alba Sorolla for preparation of samples. We thank Dr Seldin for providing CK2α′ (+/−) mice and Dr R. Morimoto for providing *C. elegans* strains. We thank Dr T. Slotkin for help with statistical analysis. This work was supported by National Institutes of Health grant R01 NS065890 to D.J.T., R01 DA031833 and R01 NS096352 to C.E., R01GM070977 to A.A., U24NS069422/U24NS078378 and R21NS083365 to C.A.R., a Holland Trice Scholar Award to C.E. and D.J.T., NIH Pre-doctoral Fellowship F31GM119375 to E.T.B. and a Postdoctoral Fellowship from the Huntington's Disease Society of America to R.G.P.

## Authors contributions

R.G.-P., E.T.B., D.W.N., A.M.J., A.D., S.U.M. and S.S.A. contributed to the study conception, design, conducted experiments and participated in data interpretation. E.C. provided human protein extracts and participated in data analysis. D.C.L., A.A., C.A.R., C.E. and D.J.T. contributed to the conception and design of experiments and data interpretation. All authors contributed to writing and editing the manuscript.

## Additional information

**Competing financial interests:** D.J.T. is a founder of Chaperone Therapeutics and a member of the SAB.

