## [Peer Review File · Nature Communications]

Reviewers' comments:

Reviewer #1 (Remarks to the Author):

In this paper, the authors found reduction of HSF1 protein and induction of its phosphorylation at S303 and S307 in pathogenic Htt-Q74-expressing PC12 cells in which heat shock-induced HSP expression was impaired. The HSF1 protein reduction and S303 phosphorylation were also observed in HD model mice brains and HD patient brains. The HSF1 protein reduction was suggested to be mediated by its phosphorylation-dependent interaction with an E3 ligase FBXW7 and proteasomal degradation. They then performed kinase inhibitor yeast screening and identified CK2 as a potent kinase phosphorylating S303/307 of HSF1. Modulation of CK2 activity altered expression levels of HSF1 and HSP70 in Htt-Q74-expressing cells. A regulatory subunit of CK2 was upregulated in HD model mice brains and HD patient brains. Finally, they showed that its heterozygous knockout suppressed HSF1 downregulation, Htt aggregation and synapse/spine abnormalities in HD model mice brains.

These observations are interesting and may provide novel aspects of HD pathogenesis. However, a number of points should be clarified for further justification. These are listed below.

Major points:

1. In all of in vitro experiments using HD cellular models, the authors treated the cells with heat-shock to check HSF1 activity and HSF1-mediated HSPs expression. This may be due to very faint activity of HSF1 in normal, unstressed condition. Does this mean the HSF1 degradation system they provided here is only functional in stressed condition? If so, it is hard to imagine that the system is actually contributing to downregulation of basal expression of HSPs in HD brains because HSF1 activity should be low in unstressed neurons. To clarify this point, the authors should check whether HSF1 is really active in the brain and altered in HD brain by EMSA or chromatin immunoprecipitation.
2. Related to the above, in the scheme (Figure 8) the authors showed that CK2 phosphorylation and FBXW7 binding are specific to active, trimeric HSF1 but not to inactive, monomeric HSF1. Is there any data support this model? Also, does mutant Htt alter HSF1 trimerization and activity?
3. In Figure 4, the authors showed that suppression of CK2 activity induced HSP upregulation in Htt-Q74-expressing cells. However, there is no data supporting that HSP upregulation is mediated through increase in HSF1 protein level. Because CK2 is a multifunctional protein and phosphorylates many types of proteins leading to HSP upregulation, direct involvement of HSF1 should be clarified, e.g. its further knockdown suppresses HSP upregulation and reduction of Htt aggregation.
4. In Figure 6B-D, the data of control mice (WT and CK2 hetero) should be added to clarify how much the expression are rescued by CK2 knockout. In Figure 6E and F, quantified data should be added. The labels of long exp and short exp seem to be not correct.
5. In Figure 7A, the spine morphology seems to be not well rescued by CK2 knockout in KI175 mice. Spine classification from immature to mature is obscure. Addition of another mice data such as locomotion, clasping or lifespan may strengthen the involvement of CK2 on HD symptoms.
6. Quantified data should be added for Figure 1F. Western blot data of HSF1 phosphorylation should be added for Figure 1E, G.
7. Regarding the quantification of the western blot for human materials (Fig.5F, Fig.S2), it seems that all controls are set to be 1.0. However, in the same gel, controls also have variation and should be normalized to certain case such as C1.
8. For human materials, especially for HD, the stage of the pathology and postmortem period

should be described. The authors said that CK2a is highly abundant in HD, but did not show which cells express CK2a. In HD, the striatum showed severe neurodegeneration and gliosis, so which cell express CK2a should be shown.

9. There are several papers describing the regulation of HSPs by mutant Htt (J Neurosci. 2007 Jan 24;27(4):868-80., EMBO J. 2008 Mar 19;27(6):827-39.). The authors should discuss the relationship between their findings and previous observations.

Minor points:

There are several typing errors, e.g. Fig 2F is 2K in the line 282. The legend for Fig.5, (F) seems to be lacking.

Reviewer #2 (Remarks to the Author):

In this study from the Thiele lab, the authors examine the role of HSF1 dysregulation in htt protein turnover and disease pathogenesis. The authors begin by convincingly documenting reduced HSF1 expression in different in vitro HD models, in HD mice, and importantly in human HD post-mortem brain. The focus then shifts to the basis for altered HSF1 metabolism, and through a phospho-proteomics analysis of transfected HSF1 in null cells, with either normal or polyQ-expanded htt, a few candidate phosphorylation sites are revealed, including S303 and S307. Altered phosphorylation in HD models is correlated with HSF1 catabolism, and then through experiments and screening in yeast, CK2-alpha' is uncovered as the likely kinase. This is validated pharmacologically and genetically, and CK2-alpha' expression increases are found in HD models, mice, and patients - this is all very compelling. The manuscript closes with a key experiment - crossing of zQ175 HD mice with CK2-alpha' k.o. mice, and documentation that heterozygous null CK2-alpha' status is sufficient to rescue systemic and neurodegenerative phenotypes in the HD mice. All in all, this is an impressive piece of work that clearly was pursued over the course of years, and nicely integrates in vitro HD model analysis with evaluation of human material, ending with validation of the thesis in a genetic cross in a full-length htt-expressing mouse model. The manuscript represents an advance for the field, and clarifies a current misunderstanding of how HSF1 dysfunction occurs in HD. Whether and/or how HSF1 dysfunction occurs in other neurodegenerative proteionopathies should now be the focus for future work, and this is important, because therapeutic interventions targeting HSF1 could be spurred on, if this is the case.

Issues that the authors should consider to strengthen the manuscript are as follows:

1) As just noted, one important aspect of this work is the observation of HSF1 dysregulation in another polyQ disease (SBMA) and possibly in more common disorders, such as AD or PD. If the authors have evaluated S303/307 phopshorylation status for other disorders, inclusion of those findings here would be useful. But, even if not, the authors should comment on how likely a possible role for CK2-alpha' / S303/307 dysregulation might be. In other words, do they think that this dysregulation is specific for HD or not?

2) What is the authors' model for CK2-alpha' increase in HD? Since they show increased mRNA levels, it is presumably transcriptional. Since htt-N-terminal fragment is a bona fide transcription factor, does htt-polyQ repress CK2-alpha' promoter function in a reporter assay? Does htt occupy the CK2-alpha' promoter? These studies would extend the understanding of mechanism, and would be relevant to point 1 above.

3) Examination of HSF1 levels in ALS and PD post-mortem material is useful data, but why not examine pathologically affected tissue samples also (if available)? ALS spinal cord should be available and would be worth evaluating.

4) The authors explain that in 10/14 striatum samples and 5/7 cortex samples, HSF1 is reduced. Can the authors provide a similar data set for immunoblotting of FBXW7? Do FBXW7 increases and HSF1 decreases match? Same should be done for CK2-alpha' immunoblot analysis, if possible.

5) Rescue of PGC-1-alpha and its targets are noted as a beneficial outcome of CK2-alpha' dosage reduction. It might be worth measuring TFEB expression to determine if this downstream target, with implications for autophagy regulation, was modulated in the genetic cross.

Minor issue: Line 441/442 where authors states "we found that double transgenic mice (KIQ175/CK2α' +/-442) significantly reduced body weight loss" is confusing. Should re-write as "displayed significantly increased body weight in comparison to..."

Reviewer #3 (Remarks to the Author):

This is a very nice piece of work in a very competitive area of research which explores the novel idea that expression of the protein quality control components by Heat Shock Transcription Factor 1 (HSF1) ameliorates biochemical and neurobiological defects, occurring in the short term in HD models, where Htt-polyQ misfolding and aggregation leads to dysfunction and death in striatal neurons and in the periphery as consequence of progressive defect in chaperone machinery efficient functioning. Authors report that in cellular and animal models of HD and in human HD patients HSF1 is degraded, resulting in dampened expression of the protein folding and quality control machinery. The study identifies a key pathway for modulation to prevent neuronal dysfunction, cell death and muscle wasting caused by protein misfolding.

The study is well-conceived and well-executed. This reviewer is quite satisfied with the significance of this study, the care in which the study was performed, and the implications of the results for human health. The results presented are self-consistent and highly convincing, and the work appears to have been well conducted. The interpretations of the results are justified by the data. The results fit entirely within the scope of Nature, and this reviewer pending minor changes recommends acceptance and publication of this work.

Minor comments:

- One interesting aspect related to the science of heat shock protein expression and heat shock-dependent cytoprotection is that this fundamental system is emerging as target for novel therapeutic strategies, based on the ability of small druggable compounds, including natural antioxidants to induce preconditioning and, as such, induce hormetic responses. Authors should mention and discuss in an appropriate paragraph their finding related to the biology of HSF, in relation to the emerging concepts of vitagene network and hormesis and their relevance in neuroprotection (see for review: Calabrese V. et al. ARS 2010, 13:1763-811; Calabrese V. et al., 2011; Calabrese V. et al., Mol Aspects Med. 2011, 32:279-304; Calabrese V. et al., 2012-2015).

Point-by-Point Responses to Reviewers' Comments: (Reviewer comments in italics)

Reviewer #1:

In this paper, the authors found reduction of HSF1 protein and induction of its phosphorylation at S303 and S307 in pathogenic Htt-Q74-expressing PC12 cells in which heat shock-induced HSP expression was impaired. The HSF1 protein reduction and S303 phosphorylation were also observed in HD model mice brains and HD patient brains. The HSF1 protein reduction was suggested to be mediated by its phosphorylation-dependent interaction with an E3 ligase FBXW7 and proteasomal degradation. They then performed kinase inhibitor yeast screening and identified CK2 as a potent kinase phosphorylating S303/307 of HSF1. Modulation of CK2 activity altered expression levels of HSF1 and HSP70 in Htt-Q74-expressing cells. A regulatory subunit of CK2 was upregulated in HD model mice brains and HD patient brains. Finally, they showed that its heterozygous knockout suppressed HSF1 downregulation, Htt aggregation and synapse/spine abnormalities in HD model mice brains.

These observations are interesting and may provide novel aspects of HD pathogenesis. However, a number of points should be clarified for further justification. These are listed below.

Major points:

1. In all of in vitro experiments using HD cellular models, the authors treated the cells with heat-shock to check HSF1 activity and HSF1-mediated HSPs expression. This may be due to very faint activity of HSF1 in normal, unstressed condition. Does this mean the HSF1 degradation system they provided here is only functional in stressed condition? If so, it is hard to imagine that the system is actually contributing to downregulation of basal expression of HSPs in HD brains because HSF1 activity should be low in unstressed neurons. To clarify this point, the authors should check whether HSF1 is really active in the brain and altered in HD brain by EMSA or chromatin immunoprecipitation.

We appreciate this point with respect to the physiological function of HSF1 in HD cells and tissues in the absence of heat shock. In the revised manuscript we have elaborated on our discussion of this point, and included new data in the revised manuscript (and included in this response below) to directly address this excellent point.

Many previous studies using transcriptomic and DNA binding approaches demonstrate that HSF1 activates several target genes under normal, unstressed conditions and it is constitutively bound to these target genes (citations in manuscript). However, HSF1 activity is greatly induced in response to stressful conditions such as heat shock, oxidative stress or other conditions that cause protein misfolding. In the original submission we demonstrated (Supplementary Figure 1H, currently **Figure 1E** in the revised manuscript) that striatal-derived cells (STHdhQ111) from an HD mouse model showed reduced HSF1 protein levels (~30% reduction) and increased HSF1-phosphorylation at 33°C (Control conditions) compared to STHdhQ7 control cells. Therefore, HSF1 degradation occurs in the absence of an *externally imposed* stress at 33°C and is exacerbated in response to heat shock conditions. We did use Hsp70 expression induction in response to heat shock as a read out for inducible HSF1 activation. A similar result is observed in the rat PC12 cell model expressing Q74 under Dox induction in Figure 1B at 37°C, (compare lanes 5 and 7, Control

conditions, without and with Dox) where a ~30% HSF1 reduction is observed and HSF1-303 and 307 phosphorylation is increased. Figures 1E-F and Figure 1H (**Figures 1F-G** and **Figure 1I** in the revised manuscript) showed reduced expression of HSF1 in HD mouse and human HD patients, increased HSF1-S303 phosphorylation and reduced basal expression of the chaperones Hsp25 and Hsp70, all in the absence of any imposed heat shock. In the revised manuscript we have now included both a shorter and a longer exposure of Hsp25 in **Figure 1E** that demonstrates decreased basal expression in the STHdhQ111 cells at 33°C. The fact that the Q111-expressing mouse striatal neurons are experiencing accumulation and aggregation of mHtt implies a proteotoxic and stressful situation that triggers HSF1 degradation that is exacerbated in response to a heat shock stress.

To further address the Reviewer's excellent point, we conducted additional experiments directly comparing STHdhQ7 and Q111 cells at 33°C (control conditions) where we show decreased HSF1 levels and reduced basal expression of the HSF1 target gene Hsp25. These data recapitulate the HSF1 degradation observed in STHdh cells under normal conditions. We also show that the HSF1 degradation system components (CK2 α' and Fbxw7) are elevated in STHdhQ111 under unstressed conditions 33°C (See **Figure A below**). We have also performed new experiments using human HD iPSC-derived MSNs from 33Q (Control) and a 60Q (HD) containing allele individuals to better recapitulate the HSF1 degradation system in a more HD patient-relevant model (See **Figure B below**). We show that MSN derived cells from a 60Q HD patient has reduced HSF1 and Hsp70 levels and increased CK2 α' and Fbxw7 levels. In the revised manuscript these new data have been added to a new **Figure 6**, now **Figure 6A**, and **Figure B**. Taken together, these results and others shown in the manuscript demonstrate that the degradation system is actually contributing to decrease levels of HSF1 under physiological conditions in polyQ expressing cells.

Figure 6. (A) Mouse STHdh Q7 and Q111 cells were cultured at 33°C for 48h and protein samples were subjected to immunoblotting for the indicated proteins. (B) Human iPSC from 33Q (Control) and 60Q (HD) individuals were differentiated into MSNs (medium spiny neurons) using 20 ng/ml BDNF, followed by BDNF withdrawal (-) or not (+) for 48h. Protein samples were subjected to immunoblotting for the indicated proteins.

Additionally, Riva et al., 2013 (*J. Huntingtons Dis.* 2012; 1(1): 33–45) demonstrated by HSF1 ChIP-seq analysis that Poly-glutamine expanded Huntingtin dramatically alters the genome wide binding of HSF1 in mouse STHdh cells and ~115 genes are differentially bound by HSF1 between STHdhQ7 and Q111 cells under basal (33°C) conditions. Our data, coupled with this report, strongly suggests that HSF1 plays an important role in

controlling target gene expression in HD models even under conditions without an imposed heat shock stress.

To address the Reviewer's question as to whether HSF1 is active in the HD brain, we also used published transcriptomic data from 44 human HD brains compared with 36 unaffected controls (Hodges et al., *Hum Mol Genet.* 2006 15;15(6):965-77) and analyzed the expression of 661 HSF1 direct target genes described in Vihervaara et al., *Proc Natl Acad Sci U S A.* 2013 Sep 3;110(36):E3388-97). In the revised manuscript we present a new **Supplementary Data 3** with all identified HSF1 targets that showed a significant decreased in expression in the caudate of human HD patients. **Supplementary Data 3** contains several representative members of the Hsp105, Hsp90, Hsp70 and Hsp40 chaperone family as well as other HSF1 target genes with important metabolic and synaptic functions that are significantly reduced in the human HD brain. These data, and our analysis of the striatum from the HD mouse model and human HD patients showing reduced Hsp70 and Hsp25 basal protein expression (**Figure 1F,G and I and Supplementary Figure 2A-C**) demonstrate the reduced activity of HSF1 in both the mouse and human HD brain.

Finally, in a new experiment (**Figure 6B**) in the revised manuscript, we have collaborated with Dr. Christopher Ross (Johns Hopkins Univ.) to demonstrate that HSF1 levels and activity are low in human HD iPSCs differentiated to MSN-like cells, and that this parallels a dramatic increase in CK2 α' and Fbxw7 levels. Dr. Ross and his colleague, Dr. Akimov, have been added as co-authors to the revised manuscript.

2. Related to the above, in the scheme (Figure 8) the authors showed that CK2 phosphorylation and FBXW7 binding are specific to active, trimeric HSF1 but not to inactive, monomeric HSF1. Is there any data support this model? Also, does mutant Htt alter HSF1 trimerization and activity?

We agree with the Reviewer that it would be very interesting to have insights as to whether the HSF1 oligomeric state is important for CK2 phosphorylation and Fbxw7-dependent degradation. Unfortunately, our data does not provide direct evidence to elucidate the specific events at this point. As the Reviewer suggested, we agree that it is also possible that these events occur in both monomeric and trimeric HSF1. Therefore, in the revised manuscript we have modified our model (**Figure 9**) accordingly to include monomeric and trimeric HSF1.

Regarding reviewer's comment about whether mutant Htt alters HSF1 trimerization and activity, Chafekar and Duenwald, *PLoS One.* 2012;7(5):e37929 showed by cross-linking and EMSA experiments that HSF1 trimerization ability is not significantly affected in HD but the total trimer form is diminished. Despite the ability of HSF1 to trimerize in the presence of mHtt, our results demonstrate that mHtt indirectly alters HSF1 activity by increasing CK2 α' and Fbxw7, causing HSF1 phosphorylation and proteasomal degradation, consequently reducing HSF1 target gene expression.

3. In Figure 4, the authors showed that suppression of CK2 activity induced HSP upregulation in Htt-Q74-expressing cells. However, there is no data supporting that HSP upregulation is mediated through increase in HSF1 protein level. Because CK2 is a multifunctional protein and phosphorylates many types of proteins leading to HSP

upregulation, direct involvement of HSF1 should be clarified, e.g. its further knockdown suppresses HSP upregulation and reduction of Htt aggregation.

We appreciate Reviewer's comment on the importance of showing HSF1-dependence for increasing Hsp expression by suppression of CK2 activity. Although **Figure 4G** in the original manuscript shows increased viability in the presence of CK2 inhibitor TID43 in a HSF1 dependent manner, it does not show HSF1-dependent Hsp induction. To ascertain whether this is HSF1-dependent we conducted new experiments using siRNA against HSF1 for 48h following CK2 inhibition in the presence of 2 different CK2 kinase inhibitors at 1 μ M in the PC12-HttQ74 expressing cell line. The results demonstrated that increased expression of Hsp70 after CK2 inhibition under pathogenic polyQ expressing conditions occurs in a HSF1-dependent manner. These data are included as **Supplementary Figure 4G** in the revised manuscript.

Figure S4. (G) PC12-Htt-Q74 expressing cells were transfected with siRNA against HSF1 or non-targeted siRNA (Scr) for 48h. Cells were treated with two different CK2 inhibitors (CK2-11# for SB21 and CK2-12# for TID43) at 1 μ M for 6h. Cells were heat shocked at 42°C for 1h and allowed to recover at 37°C for 7h. Protein samples were collected and subjected to immunoblotting for the indicated proteins.

4. In Figure 6B-D, the data of control mice (WT and CK2 hetero) should be added to clarify how much the expression are rescued by CK2 knockout. In Figure 6E and F, quantified data should be added. The labels of long exp and short exp seem to be not correct.

In the revised manuscript the quantitative mRNA expression data for all control mice (WT and CK2^{+/-}) (n=3) and data presented in **Figure 6B-D** (now **Figure 7B-D** in the revised manuscript) has been added to **Supplementary Figure S6**, as the Reviewer has suggested. Control data for CK2 subunit expression is now **Supplementary Figure S6C**, control data for Hsp70 and Hsp25 chaperone expression is now **Supplementary Figure S6D** and control data for PGC1 α and its downstream targets is now **Supplementary Figure S6E**. We have elected to show expression data for KIQ175 and the double mutant CK2 α' +/-:KIQ175 mice, relative to WT control (now **Supplementary Figure S6C-E**), set to 1 in the revised manuscript **Figure 7B-D**, to show the differences between these two mouse models.

Figure S6. (C-E) qRT-PCR analysis for WT, KIQ175, CK2 α' (+/-) and KIQ175/ CK2 α' (+/-) mice was conducted from striatal mRNA at 6 months of age for: (C) Casein kinase 2 catalytic (α and α') and regulatory subunits (β), (D) Hsp70 and Hsp25 and (E) Mitochondrial activity-related genes PGC1 α and its downstream targets CYCs, NDUFS3 and TFAM. All data was normalized to GAPDH expression and to WT expression levels set as 1 (G) Image quantification for CK2 α' , HSF1 and Hsp70 protein levels in the striatum of WT, KIQ175, CK2 α' (+/-) and KIQ175/ CK2 α' (+/-) mice at 6 months of age (n=3).

As the Reviewer has suggested, the quantified data from n=3 mice for **Figure 6E (Revised manuscript Figure 7E)** has been added as **Supplementary Figure S6G**. Quantification of **Figure 6F** is shown below the image in the revised manuscript (Revised manuscript **Figure 7F**) and represents an average of n=2 mice shown in the figure. Labels have been modified accordingly.

5. In Figure 7A, the spine morphology seems to be not well rescued by CK2 knockout in KIQ175 mice. Spine classification from immature to mature is obscure. Addition of another mice data such as locomotion, clasping or lifespan may strengthen the involvement of CK2 on HD symptoms.

We thank the Reviewer for the opportunity to clarify our description of the spine morphology analysis. Detailed and specific data analysis protocols for spine classification are provided by McKinstry et al., 2014 and Risher et al., 2014, as referenced in the manuscript. In the revised manuscript we provide additional detailed information on the methodology used to determine and classify the neuronal spines, which has been added to **Supplementary Experimental Procedures as follows:**

The classification of the different spines is based on on the width, length, and length:width ratio measurements taken using the Reconstruct software (available at <http://synapses.cim.utexas.edu>; RRID:nif-0000-23420) designed and validated by Risher et al., 2014. To identify and classify the spines we chose 10 μ m segments of dendrites. Spines were identified on selected dendritic stretches. z-length (spine length) and spine head width were measured for each spine. These measurements were exported to Microsoft Excel. A custom Excel macro was used to classify spines based on the width, length, and length:width ratio measurements taken in Reconstruct. Spines were categorized based on the following hierarchal criteria: (1) more than one spine head = "branched spine," (2) head width >0.7 μ m = "mushroom spine," (3) length > 2 μ m = "filopodia," (4) length:width >1 =

“thin spine,” and (5) $\text{length:width} \leq 1 =$ “stubby spine.” Branched and mushroom spines were identified as Mature spines, thin and stubby spines were categorized as intermediate spines, and filopodia were classified as immature spines. Statistical analyses of changes in spine density, length, width, and spine type were conducted in the Statistica program (StatSoft): A total of 3 animals/genotype, 15 dendrites/animal, 45 dendrites per genotype total were analyzed in MSNs in the dorsal striatum. The number of spines analyzed per neuron type per age per genotype exceeded 900.

Given the automated analysis described to identify and classify the different spine types and numbers per neuron and per genotype, we are confident that the spine morphology is rescued by CK2 α' knockout in the KIQ175 mice. The data and statistical analysis are provided in **Figure 8B** in the revised manuscript. In response to the Reviewer’s point, in the revised manuscript we have also selected a different picture for **Figure 8A** to better represent the spine morphology rescued in the KIQ175: CK2 $\alpha'^{+/-}$ mice quantified in **Figure 8B**.

We agree with the Reviewer that additional physiological and/or behavioral data such as locomotion, clasping or lifespan may strengthen our observations. However to perform these analyses at a late onset would require a large amount of time to breed and obtain a sufficient number of animals representing all of the relevant genotypes. We believe that body weight analysis is a well described HD phenotype and our analysis showed that CK2 α' (+/-):KIQ175 had increased body weight compared to the KIQ175 HD model. In the revised manuscript we include a new longitudinal body weight measurement during 9 weeks after weaning, presented in **Supplementary Figure S7C**, that shows increased body weight of the CK2 α' (+/-):KIQ175 compared to KIQ175 over this time period. These results and the improvement observed in MSN spine maturation (**Figure 8A, B**) and in thalamo-striatal synapse activity (**Figure 8D, F**) strongly suggest the involvement of CK2 α' in driving some of the HD symptoms.

Figure S7. (C) Body weight longitudinal study for WT (n=4), KIQ175 (n=3), CK2 α' (+/-) (n=5) and KIQ175/ CK2 α' (+/-) (n=3) over 9 weeks following weaning, represented in grams (g). Only males were used in this study to minimize weight differences.

6. Quantified data should be added for Figure 1F. Western blot data of HSF1 phosphorylation should be added for Figure 1E, G.

In the revised manuscript we have quantified a total of 3 pictures per genotype (n=3) and average data and statistical analysis has been added to **Supplementary Figure 1**, now **Figure S11**. We have analyzed the amount of cells that stained positive for HSF1 and calculate the percentage of cells with respect to the total amount of cells stained with DAPI (nuclear marker). HSF1 Ser-303 Phosphorylation was quantified as the ratio between the

number of red positive cells (staining for HSF1) and the number of purple positive cells (staining for HSF1-S303-P).

Figure S1. (I) Image quantification data of IHC experiments shown in **Figure 1G**. Left panel shows the % of HSF1 positive cells normalized to the total number of cells stained by DAPI in the WT and KI Q175 mice at 12 months of age in the dorsal striatum. Right panel shows the ratio between HSF1 positive cells and HSF1-S303-P stained cells. Data was normalized to WT levels set as 1 for n=3 independent experiments. (*p-value < 0.05).

Although we would like to include western blot data of HSF1 phosphorylation for **Figures 1F and 1H**, the quality of anti-HSF1-S303P signal using mouse tissues in western blotting is very poor. Therefore, we show HSF1-S303P by immunohistochemistry (**Figure 1G in the revised manuscript**) as well as HSF1 phosphorylation in mouse derived striatal cells STHdhQ7 and Q111 where we demonstrate increased HSF1 S303P and S307P in the STHdhQ111 cell line compared to Q7 cells (**Figure 1E in the revised manuscript**).

7. Regarding the quantification of the western blot for human materials (Fig.5F, Fig.S2), it seems that all controls are set to be 1.0. However, in the same gel, controls also have variation and should be normalized to certain case such as C1.

We used age- and sex-matched pairs for control and HD patients, where every pair has different characteristics regarding age, sex and postmortem tissue harvest time (see **Supplementary Table 1**). These properties make it very difficult to normalize all data to just one arbitrary control sample. Therefore, we consider it appropriate to compare every HD case to its corresponding control, with similar properties, that is set to 1.

8. For human materials, especially for HD, the stage of the pathology and postmortem period should be described. The authors said that CK2a is highly abundant in HD, but did not show which cells express CK2a. In HD, the striatum showed severe neurodegeneration and gliosis, so which cell express CK2a should be shown.

In the revised manuscript the pathology stage and postmortem period for all human HD samples have been included in **Supplementary Table 1**.

We appreciate Reviewer's suggestion on the importance of showing which cells express CK2 α' . To address this question we have performed several IHC experiments in the dorsal striatum of WT and KI Q175 mice at 6 months of age, where we show that CK2 α' protein levels are increased in the KI Q175 mice (**Figure 5C**). We first validated the specificity of the CK2 α' antibody by using WT and CK2 α' (-/-) mice brain slices where no signal was observed for the CK2 α' (-/-) mice (**new Supplementary Figure S5A**). To assess which cell

types express high levels of CK2 α' in the KIQ175 mice, we conducted IHC experiments for several makers; NeuN (neurons), Darpp-32, Ctip2 and Fox1P (Medium Spiny Neurons), GS (astrocytes) and CD68 (reactive microglia). Our new data demonstrates that in WT mice the basal expression of CK2 α' co-localizes with NeuN, Darpp-32, Ctip2 and Fox1P suggesting that MSNs are expressing CK2 α' . In the KIQ175 mice we observed that CK2 α' expression is elevated in the dorsal striatum by IHC, recapitulating the data shown in **Figure 5C**. We also observed that Darpp-32 positive cells were significantly low in the KIQ175 mice, recapitulating the decreased mRNA levels encoding this protein as previously described by other authors in this and other HD mouse models (Luthi-Carter et al., *Hum Mol Genet.* 2000 22;9(9):1259-71; Menalled et al., *PLoS One.* 2012;7(12):e49838). We observed co-localization of CK2 α' with the few cells that still express Darpp-32 in the KIQ175 mice. However, a great number of cells expressing CK2 α' in the KIQ175 mice did not co-express Darpp-32. By using NeuN staining for neurons and the alternative Ctip2 and Fox1p markers for MSNs (Arlotta et al., *J Neurosci.* 2008; 16;28(3):622-32), we demonstrated that the increased CK2 α' signal co-localized with all 3 markers in KIQ175 mouse striatum, indicating that MSNs are responsible for the increase in CK2 α' in HD. However, our data suggests that MSNs that have increased CK2 α' expression in the KIQ175 mouse HD model have severely diminished Darpp-32 expression.

The use of GS and CD68 demonstrated that CK2 α' does not co-localize with neither astrocytes nor reactive microglia. Interestingly, we observed that despite no co-localization with these two sub-types of cells, MSNs that have elevated CK2 α' expression in the KIQ175 mouse were in close proximity to these cells, suggesting that they may be targeted for clearance. Additional control experiments using the 1C2 antibody that recognized mutant Htt shows the expression of mHtt in the KIQ175 MSNs and it does co-localized with CK2 α' in the KIQ175 striatum. Together, the new data that we provide in the revised manuscript demonstrates that increased CK2 α' expression occurs in MSNs in the striatum of KIQ175 HD mice.

Data showing co-expression of CK2 α' with the MSNs markers Ctip2 and Fox1p has been included in **Figure 5E** in the revised manuscript. IHC experiments showing CK2 α' specificity in WT and CK2 α' (-/-) mice is shown in **Supplementary Figure 5A** in the revised manuscript. Data showing CK2 α' , NeuN and CD68 has been included in **Supplementary Figure 5B**. Data showing CK2 α' and Darpp-32 has been included in **Supplementary Figure 5C**. Data showing CK2 α' and Glutamine synthetase (GS) has been included in **Supplementary Figure 5D** and data showing CK2 α' and mHtt has been included in **Supplementary Figure 5E**.

Figure 5. (E) Coronal section of the striatum of WT and KIQ175 mice at 6 months of age, showing co-localization of CK2 α' (red) with Ctip2 (green) and Fox1p (magenta) labeled MSN in the merged image. Scale bar: 10 μ m.

Figure S5. (A) Coronal section of the striatum of WT and CK2 α' (-/-) mice at 3 months of age showing specific staining of CK2 α' only in WT mice. (B-E) Coronal section of the striatum of WT and KIQ175 mice at 6 months of age, showing staining of (B) CK2 α' (red) and the neuronal marker NeuN (magenta) co-localize but there is no co-localization with the reactive microglia marker CD68 (green). (C) Co-localization of CK2 α' (red) and the Darpp32 (green) MSN marker. (D) CK2 α' (red) and GS, astrocytes (green) do not co-localize. (E) CK2 α' (red) and mHtt (green) co-localize in the KIQ175 mice. Nuclei were detected with DAPI. Scale bar: A 40 μ m, B-E 10 μ m.

We agree with the reviewer that in HD, the striatum shows severe neurodegeneration and gliosis (Sorolla et al., *Free Radic Biol Med.* 2008 1;45(5):667-78). However, it has been described that such degeneration is less severe in the brain of HD mice where neither gross neuronal loss nor gliosis has been reported (Mangiarini et al., *Cell.* 1996 1;87(3):493-506.; Heikkinen et al., *PLoS One* 2012;7(12):e50717). Transcriptomic analysis in the R6/2 mice and KIQ175 mice have revealed that the decreased expression of striatal neuronal signaling genes such as Darpp-32, is not due to large shifts in striatal cell populations (Luhi-Carter et al., *Hum Mol Genet.* 2000 22;9(9):1259-71; Menalled et al., *PLoS One.* 2012;7(12):e49838) but rather suggests a signature in these neurons due to the presence of mHtt. In addition, estimated neuronal density and neuronal number by stereologic analysis in heterozygous KIQ175 mice at 10 months shows no significant

changes compared to WT mice, despite the observed HD phenotype in those mice (Heikkinen et al., *PLoS One*. 2012;7(12):e50717). These data support our findings that despite the decrease in Darpp-32 levels in MSNs, the amount of MSNs in the KIQ175 model is similar to WT mice (detected by Ctip2 and Fox1p) but they show increased expression of CK2 α '.

9. *There are several papers describing the regulation of HSPs by mutant Htt (J Neurosci. 2007 Jan 24;27(4):868-80., EMBO J. 2008 Mar 19;27(6):827-39.). The authors should discuss the relationship between their findings and previous observations.*

We appreciate Reviewer pointing out these two interesting papers and we have referenced and discussed these publications in the revised manuscript in lines 474-483.

J Neurosci. 2007 Jan 24;27(4):868-80. The induction levels of heat shock protein 70 differentiate the vulnerabilities to mutant huntingtin among neuronal subtypes. Tagawa K¹, Marubuchi S, Qi ML, Enokido Y, Tamura T, Inagaki R, Murata M, Kanazawa I, Wanker EE, Okazawa H.

EMBO J. 2008 Mar 19;27(6):827-39. doi: 10.1038/emboj.2008.23. Epub 2008 Feb 21. Mutant Huntingtin reduces HSP70 expression through the sequestration of NF-Y transcription factor. Yamanaka T¹, Miyazaki H, Oyama F, Kurosawa M, Washizu C, Doi H, Nukina N.

Minor points:

There are several typing errors, e.g. Fig 2F is 2K in the line 282. The legend for Fig.5, (F) seems to be lacking.

We thank the Reviewer for pointing out typos, which have been corrected in the revised manuscript. The legend **for Fig. 5** has also been modified in the revised manuscript.

Reviewer #2 (Remarks to the Author):

In this study from the Thiele lab, the authors examine the role of HSF1 dysregulation in htt protein turnover and disease pathogenesis. The authors begin by convincingly documenting reduced HSF1 expression in different in vitro HD models, in HD mice, and importantly in human HD post-mortem brain. The focus then shifts to the basis for altered HSF1 metabolism, and through a phospho-proteomics analysis of transfected HSF1 in null cells, with either normal or polyQ-expanded htt, a few candidate phosphorylation sites are revealed, including S303 and S307. Altered phosphorylation in HD models is correlated with HSF1 catabolism, and then through experiments and screening in yeast, CK2-alpha' is uncovered as the likely kinase. This is validated pharmacologically and genetically, and CK2-alpha' expression increases are found in HD models, mice, and patients - this is all

very compelling. The manuscript closes with a key experiment - crossing of zQ175 HD mice with CK2-alpha' k.o. mice, and documentation that heterozygous null CK2-alpha' status is sufficient to rescue systemic and neurodegenerative phenotypes in the HD mice. All in all, this is an impressive piece of work that clearly was pursued over the course of years, and nicely integrates in vitro HD model analysis with evaluation of human material, ending with validation of the thesis in a genetic cross in a full-length htt-expressing mouse model. The manuscript represents an advance for the field, and clarifies a current misunderstanding of how HSF1 dysfunction occurs in HD. Whether and/or how HSF1 dysfunction occurs in other neurodegenerative proteionopathies should now be the focus for future work, and this is important, because therapeutic interventions targeting HSF1 could be spurred on, if this is the case.

Issues that the authors should consider to strengthen the manuscript are as follows:

1) As just noted, one important aspect of this work is the observation of HSF1 dysregulation in another polyQ disease (SBMA) and possibly in more common disorders, such as AD or PD. If the authors have evaluated S303/307 phosphorylation status for other disorders, inclusion of those findings here would be useful. But, even if not, the authors should comment on how likely a possible role for CK2-alpha' / S303/307 dysregulation might be. In other words, do they think that this dysregulation is specific for HD or not?

We agree with the Reviewer that it would be very interesting to ascertain whether the HSF1 degradation mechanism that we describe here in HD is conserved in other neurodegenerative diseases. We comment and reference in the revised manuscript that HSF1 protein depletion has also been reported for other neurodegenerative diseases such as AD, PD and ALS (Kondo et al., Nat Commun. 2013;4:1405.; Jiang et al., Brain Res. 2013 Jun 26;1519:105-11; Kim et al., Hum Mol Genet. 2016 Jan 15;25(2):211-22) and it has also been described that over-expression of CK2 and CK2 hyper-activation is linked to these neurodegenerative diseases (Masliah et al., Am J Pathol. 1992 140(2):263-8.; Rosenberger et al., J Neuroinflammation. 2016 Jan 6;13:4). We have not evaluated this possibility in the work described in this manuscript so that we can focus on a more complete characterization of HSF1 degradation, its mechanism and its consequences in HD.

2) What is the authors' model for CK2-alpha' increase in HD? Since they show increased mRNA levels, it is presumably transcriptional. Since htt-N-terminal fragment is a bona fide transcription factor, does htt-polyQ repress CK2-alpha' promoter function in a reporter assay? Does htt occupy the CK2-alpha' promoter? These studies would extend the understanding of mechanism, and would be relevant to point 1 above.

Understanding the mechanisms that control increases in CK2 α' and Fbxw7 in HD is a very important question and is an ongoing investigation in our lab. As the Reviewer suggests, it is possible that htt-polyQ is involved in the direct transcriptional up-regulation of Ck2 α' and or Fbxw7. However, we have not yet explored whether CK2 α' and Fbxw7 increased expression is due to altered transcriptional activity of any particular transcription factor or if there are different post-transcriptional events that may stabilize and increase their expression. Stimulated by the Reviewer's comments and due to the absence of any mechanistic data provided in **Supplementary Figure 1I** from the initial submission, we

have removed it from the revised manuscript. Among our current studies, we will focus on this important question.

3) Examination of HSF1 levels in ALS and PD post-mortem material is useful data, but why not examine pathologically affected tissue samples also (if available)? ALS spinal cord should be available and would be worth evaluating.

Our intention of analyzing striatum of ALS and PD (although they are not the primary affected tissue) was to compare neuronal tissue from other diseases to that affected in HD to ascertain the specificity of HSF1 depletion in the striatum of Huntington's disease. We agree with the Reviewer's suggestion that testing affected tissue samples for ALS or PD would be interesting experiments. These studies have recently been reported by others (Kim, E., et al. *Hum. Mol. Genet.* (2015) 25(2):211-22) showing that HSF1 is depleted in the inferior parietal lobe of PD patients, the most affected area in PD. On the other hand, Pei-Yi Lin et al., (*Molecular Neurodegeneration* 20138:43 DOI: 10.1186/1750-1326-8-43) have demonstrated that HSF1 protein levels are not affected in the spinal cord of an SOD1-H46R/H48Q ALS mouse model. Given the fact that affected tissue of ALS and PD have been already evaluated by other author's and that our data presented in **Figure 1** in the original submission does not represent affected tissue, we have elected to remove these data from the revised manuscript. Accordingly, **Table 2** and **Supplementary Figure S2D** have also been removed from Supplementary Information.

4) The authors explain that in 10/14 striatum samples and 5/7 cortex samples, HSF1 is reduced. Can the authors provide a similar data set for immunoblotting of FBXW7? Do FBXW7 increases and HSF1 decreases match? Same should be done for CK2-alpha' immunoblot analysis, if possible.

To address this excellent suggestion, we performed new western blotting analysis of CK2 α' and Fbxw7 for all 14 available human striatum samples to test the correlation between HSF1, CK2 α' and Fbxw7 levels and we quantified CK2 α' and Fbxw7 relative to GAPDH protein levels using image J. In the revised manuscript we provide a new **Supplementary Figure 2A and 2B** and image quantification analysis to show the relative fold change of HSF1, CK2 α' and Fbxw7 protein levels for all 14 HD patients. We observed that 9 out of 14 HD patients showed increased CK2 α' expression in the striatum compared to control samples and 6 out of 14 showed increased Fbxw7. Among all 10 patients that showed reduced HSF1, 7 HD patients showed increased CK2 α' and 5 out of those 10 patients showed Fbxw7 increase.

We also note the addition of new data (**Figure 6B**) in the revised manuscript, in which we have collaborated with Dr. Christopher Ross (Johns Hopkins Univ.) to demonstrate that HSF1 levels and activity are low in human HD iPSCs differentiated to MSN-like cells, and that this parallels a dramatic increase in CK2 α' and Fbxw7 levels. Dr. Ross and his colleague, Dr. Akimov, have been added as co-authors to the revised manuscript.

We also noticed a mistake in the patient labeling in **Figure 2G** in the original manuscript and it has been corrected in the revised manuscript.

Figure S2. (A, B) CK2α' and Fbxw7 bands from immunoblots were quantified using Image J and the protein values were normalized using GAPDH as loading control and referenced to the corresponding age- and sex-matched control patient set to 1.

5) *Rescue of PGC-1-alpha and its targets are noted as a beneficial outcome of CK2-alpha' dosage reduction. It might be worth measuring TFEB expression to determine if this downstream target, with implications for autophagy regulation, was modulated in the genetic cross.*

We agree with the Reviewer's suggestion that testing other PGC1α target gene expression, such as the TFEB transcription factor, could strengthen the rationale for the beneficial outcome of the CK2α' dosage reduction experiment. Therefore, inspired by the Reviewer's suggestion, we performed qRT-PCR analysis for TFEB expression using primers: Fw-ACAGTCTCCGTTCCATCACC and Rv:GGCCTCAGGAGACATGGTAG for WT, KI175, CK2α' +/- and KI175: CK2α' +/- in the striatum of 6 month old mice. While the expression of TFEB has been shown to be reduced in the striatum of 13 week old N171-82Q HD mice and rescued by PGC1α overexpression (Tsunemi et al., *Science Translational Medicine* 11 Jul 2012:Vol. 4, Issue 142, pp. 142ra97DOI: 10.1126/scitranslmed.3003799), there is no available data for the new KI175 HD mouse model used in our study. Our analysis has revealed that TFEB expression is not reduced in the KI175 mice compared to WT littermates at 6 months of age, when disease phenotypes are observed, and we do not see TFEB expression alteration when we compared KI175 and KI175: CK2α' +/- mice. The discrepancy with respect to TFEB expression between the N171-82Q model and the KI175 model may be due to the different mHtt constructs expressed in each model and the different severity of these two HD mouse models, with a 5-6 month life expectancy for the N171-82Q mouse and >12 months for the KI175.

Minor issue: Line 441/442 where authors states "we found that double transgenic mice (KI175/CK2α' +/-442) significantly reduced body weight loss" is confusing. Should re-write as "displayed significantly increased body weight in comparison to..."

We have modified this sentence according to the Reviewer's suggestion in the revised manuscript.

Reviewer #3 (Remarks to the Author):

This is a very nice piece of work in a very competitive area of research which explores the novel idea that expression of the protein quality control components by Heat Shock Transcription Factor 1 (HSF1) ameliorates biochemical and neurobiological defects, occurring in the short term in HD models, where Htt-polyQ misfolding and aggregation leads to dysfunction and death in striatal neurons and in the periphery as consequence of progressive defect in chaperone machinery efficient functioning. Authors report that in cellular and animal models of HD and in human HD patients HSF1 is degraded, resulting in dampened expression of the protein folding and quality control machinery. The study identifies a key pathway for modulation to prevent neuronal dysfunction, cell death and muscle wasting caused by protein misfolding.

The study is well-conceived and well-executed. This reviewer is quite satisfied with the significance of this study, the care in which the study was performed, and the implications of the results for human health. The results presented are self-consistent and highly convincing, and the work appears to have been well conducted. The interpretations of the results are justified by the data. The results fit entirely within the scope of Nature, and this reviewer pending minor changes recommends acceptance and publication of this work.

Minor comments:

- One interesting aspect related to the science of heat shock protein expression and heat shock-dependent cytoprotection is that this fundamental system is emerging as target for novel therapeutic strategies, based on the ability of small druggable compounds, including natural antioxidants to induce preconditioning and, as such, induce hormetic responses. Authors should mention and discuss in an appropriate paragraph their finding related to the biology of HSF, in relation to the emerging concepts of vitagene network and hormesis and their relevance in neuroprotection (see for review: Calabrese V. et al. ARS 2010, 13:1763-811; Calabrese V. et al., 2011; Calabrese V. et al., Mol Aspects Med. 2011, 32:279-304; Calabrese V. et al., 2012-2015).

We sincerely appreciate the Reviewer's suggestion. In the revised manuscript we have discussed the interesting topics of hormesis and the vitagene network with respect to neuroprotection in lines 497-499. Due to reference number limitation we have included only one of the proposed references by the Reviewer: Calabrese V. et al., Mol Aspects Med. 2011, 32:279-304

REVIEWERS' COMMENTS:

Reviewer #1 (Remarks to the Author):

The authors responded well to the reviewers' comments, except the answer for the comment 7 in the responses to reviewer #1. For this reviewer, it is appropriate to show the variation among controls, but the readers should evaluate it.

Reviewer #2 (Remarks to the Author):

The authors have been responsive to my concerns and have adequately addressed them. Furthermore, the manuscript contains a number of significant new experimental results, including especially the analysis of the HD patient stem cell model. All in all, the work has been substantively improved. My only remaining suggestion is that the authors may wish to revise the title, as it is a bit unwieldy and indirect.

Reviewer #3 (Remarks to the Author):

The paper is now acceptable for publication on Nature

Point-by-Point Response to Reviewer comments

Reviewer 1:

The authors responded well to the reviewers' comments, except the answer for the comment 7 in the responses to reviewer #1. For this reviewer, it is appropriate to show the variation among controls, but the readers should evaluate it.

We have modified Figure S2, sections C, D, E and G to quantitate protein levels for HSF1, CK2alpha prime and Fbxw7 in aggregate to show the variation between control and HD patients to address this concern.

Reviewer 2:

The authors have been responsive to my concerns and have adequately addressed them. Furthermore, the manuscript contains a number of significant new experimental results, including especially the analysis of the HD patient stem cell model. All in all, the work has been substantively improved. My only remaining suggestion is that the authors may wish to revise the title, as it is a bit unwieldy and indirect.

Honestly, in response to this Reviewer we considered other title options, but we feel that the original title is the most descriptive of what our manuscript shows and its implications in HD.

Reviewer 3:

The paper is now acceptable for publication on Nature